# Towards Foundation Models for Learning on Tabular Data

## Abstract

Learning on tabular data underpins numerous real-world applications. Despite considerable efforts in developing effective learning models for tabular data, current transferable tabular models remain in their infancy, limited by either the lack of support for direct instruction following in new tasks or the neglect of acquiring foundational knowledge and capabilities from diverse tabular datasets. In this paper, we propose Tabular Foundation Models (TabFMs) to overcome these limitations. TabFMs harness the potential of generative tabular learning, employing a pre-trained large language model (LLM) as the base model and fine-tuning it using purpose-designed objectives on an extensive range of tabular datasets. This approach endows TabFMs with a profound understanding and universal capabilities essential for learning on tabular data. Our evaluations underscore TabFM's effectiveness: not only does it significantly excel in instruction-following tasks like zero-shot and in-context inference, but it also showcases performance that approaches, and in instances, even transcends, the renowned yet mysterious closed-source LLMs like GPT-4. Furthermore, when fine-tuning with scarce data, our model achieves remarkable efficiency and maintains competitive performance with abundant training data. Finally, while our results are promising, we also delve into TabFM's limitations and potential opportunities, aiming to stimulate and expedite future research on developing more potent TabFMs.[1]

## 1 Introduction

Tabular data is ubiquitous in numerous critical industrial domains, including but not limited to healthcare, finance, retail, sustainability, and climate, due to the widespread adoption of relational databases in these domains (Shwartz-Ziv & Armon, 2022b; Hegselmann et al., 2023; Levin et al., 2023; Vogel et al., 2023). As a result, learning on tabular data serves as the foundation for a vast array of real-world applications, such as disease risk stratification, credit assessment, and sales volume prediction, thereby attracting broad research attention from the machine learning community.

Over the last decade, researchers have made tremendous efforts to develop effective learning algorithms for tabular data. Initially, the focus was on designing effective learning models with predefined data schemas and prediction tasks, such as gradient boosting decision trees (Chen & Guestrin, 2016; Ke et al., 2017; Prokhorenkova et al., 2018) and neural networks specifically tailored for tabular deep learning (Huang et al., 2020; Arik & Pfister, 2021; Katzir et al., 2021; Gorishniy et al., 2021). In recent years, motivated by the remarkable success of foundation models in language and vision data (Brown et al., 2020; Radford et al., 2021), there has been growing interest in developing *transferable* tabular models that can be easily adapted to a wide variety of downstream tasks. Two prominent paradigms have emerged: one that designs specific neural architectures to enable transfer learning on tabular data (Wang & Sun, 2022; Levin et al., 2023; Zhu et al., 2023), and another that adapts large language models (LLMs) to learn over tabular data (Dinh et al., 2022; Hegselmann et al., 2023; Wang et al., 2023; Vogel et al., 2023), both of which can be viewed as endeavors to build foundation models on tabular data.

However, existing efforts towards tabular foundation models are still in their infancy. First, methods in the first paradigm, developing neural architectures with cross-table transferability, predominantly

---

[1]Data and code will be open sourced.

require a fine-tuning stage for each new task, which lags behind language and vision foundation models in instruction following (no tuning needed). While the second paradigm can inherit some instruction-following capabilities from pretrained LLMs, existing methods in this paradigm primarily focus on the independent adaptation of LLMs to each tabular dataset (task). For example, Dinh et al. (2022); Hegselmann et al. (2023) fine-tune pretrained LLMs on multiple tabular datasets separately, and Zhang et al. (2023); Wang et al. (2023) leverage LLMs to augment training data for a target task. This limitation, overlooking the opportunity to learn foundational knowledge and capabilities from diversified datasets and tasks, also contradicts the successful experiences of foundation models in language and vision data.

To address these limitations, we propose *Tabular Foundation Models* (TabFMs) that 1) support both fine-tuning and instruction following (no tuning needed) for new tasks, and 2) acquire foundational knowledge and universal capabilities from diverse tabular datasets across different domains. To achieve the first goal, we leverage a pretrained LLM as the base model to inherit general knowledge and basic instruction-following capabilities, similar to Dinh et al. (2022); Hegselmann et al. (2023). However, directly utilizing LLMs does not tackle the challenge of realizing the second goal because pretrained LLMs possess general knowledge learned from large-scale text corpora, whereas learning over tabular data additionally demands data-driven knowledge that involves a comprehensive understanding of numerical information and statistical patterns. To bridge the gap between general knowledge learned from text corpora and data-driven knowledge required for learning on tabular data, we develop *generative tabular learning* with specifically designed objectives for LLMs, entailing a fine-tuning process on a wide range of tabular datasets to stimulate the acquisition of foundational knowledge and universal capabilities essential for understanding tabular data. This *generative tabular learning* equips the resulting TabFM with significantly enhanced capabilities in both instruction following and task-specific fine-tuning for new tabular learning tasks.

In our experiments, we employ LLaMA 2 (Touvron et al., 2023) as the base LLM, construct a large-scale data collection containing 115 public tabular datasets for generative tabular learning, and evaluate our approach on nine datasets (non-overlapping with our 115 datasets) built by Hegselmann et al. (2023). Concerning instruction following, our results show that our approach substantially improves the base LLaMA model and GPT-3.5 (Brown et al., 2020), outperforms TabLLM (Hegselmann et al., 2023) most of the time, and surprisingly, approaches or occasionally surpasses GPT-4 (OpenAI, 2023). In terms of fine-tuning for new tasks, our approach exhibits competitive performance compared to various state-of-the-art solutions for tabular learning. These results clearly demonstrate the significance of our approach in transforming LLMs into TabFMs. Moreover, we provide an in-depth discussion on our failure cases, limitations of the existing implementation, remaining challenges, and potential opportunities to explore.

We hope our findings and insights can inspire and expedite future research on developing more powerful foundation models for tabular data. In summary, our contributions include:

- We explicitly formalize two critical aspects for TabFMs: supporting both fine-tuning and instruction following for new tasks, and acquiring foundational knowledge and universal capabilities to empower tabular learning across diverse domains.
- We devise generative tabular learning to transform LLMs into TabFMs, identifying potential directions towards foundation models for learning on tabular data.
- We have constructed a large-scale data collection for research and included comprehensive comparisons with state-of-the-art solutions for tabular learning as well as renowned LLMs.

## 2 RELATED WORK

Our literature review encompasses four aspects.

**Learning on Tabular Data** The development of effective learning algorithms for tabular data has been a longstanding research topic. In the early days, tree-based models were found to be particularly suitable for tabular data, leading to the development of several gradient boosting decision trees (Chen & Guestrin, 2016; Ke et al., 2017; Prokhorenkova et al., 2018). Subsequently, as deep learning gained prominence (LeCun et al., 2015), numerous studies attempted to create suitable network architectures for tabular data (Huang et al., 2020; Arik & Pfister, 2021; Katzir

et al., 2021; Gorishniy et al., 2021) and introduced self-supervised learning schemes (Yoon et al., 2020; Somepalli et al., 2021; Ucar et al., 2021; Bahri et al., 2022). Despite these tabular neural networks not consistently outperforming tree-based models (Grinsztajn et al., 2022b; Shwartz-Ziv & Armon, 2022b; Gorishniy et al., 2021), further research has advanced tabular deep learning and established new state-of-the-art results in few-shot (Nam et al., 2023; Hollmann et al., 2023) and transfer learning scenarios (Wang & Sun, 2022; Levin et al., 2023; Zhu et al., 2023). These studies can be considered early attempts towards tabular foundation models, as they significantly improved data efficiency and aimed for easy adaptation to new tasks, which are core capabilities of foundation models (Bommasani et al., 2021). However, these methods predominantly necessitated re-training or fine-tuning to adapt to new data schemas and tasks. There are a few exceptions but having certain limitations, such as TransTab (Wang & Sun, 2022) demonstrating successful zero-shot transferring within clinical trial datasets, and TabPFN (Hollmann et al., 2023) holding in-context capabilities but not support zero-shot inference. In contrast, this paper systematically investigates zero-shot and in-context generalization in tabular deep learning, emphasizing extreme data efficiency and general transferability across diverse domain knowledge, data schemas, and tasks.

**LLM-assisted Tabular Deep Learning** The remarkable success of LLMs, scaled to unprecedented sizes and trained on massive text corpus, has demonstrated their broad knowledge and incredible capabilities in transfer learning and instruction following (Brown et al., 2020; Ouyang et al., 2022). This success has motivated the adaptation of LLMs to tabular deep learning with the aims of 1) inheriting the broad world knowledge already learned, 2) enabling instruction following to support arbitrary tasks without tuning, and 3) effectively utilizing meta-information in tabular data, such as column names, task descriptions, and background knowledge. LIFT (Dinh et al., 2022) proposed language-interfaced fine-tuning, which fine-tuned GPT-3 (Brown et al., 2020) and GPT-J (Wang & Komatsuzaki, 2021) on multiple tabular learning datasets, finding that the performance of fine-tuned LLMs was roughly comparable to traditional solutions. TabLLM (Hegselmann et al., 2023), a subsequent study adopting T0 (Sanh et al., 2022) as the base LLM, reported competitive performance of fine-tuned LLMs in very few-shot scenarios but slight underperformance compared to classical tree models (Chen & Guestrin, 2016; Ke et al., 2017) and tabular networks (Hollmann et al., 2023) when more shots were available. In contrast to these studies that directly utilize LLMs pretrained on language data, our paper proposes to first bridge the gap between the general knowledge learned by LLMs and the specific abilities required to truly understand tabular data, such as numeracy skills, capturing feature dependencies, and understanding feature distributions. To achieve this, we introduce an additional learning stage on tabular data and develop several specific objectives to stimulate the learning of crucial abilities in understanding tabular data. Additionally, AnyPredict (Wang et al., 2023) also employed LLMs to build supplementary data for a specific target task, and TapTap (Zhang et al., 2023) used LLMs to generate synthetic tabular data, both of which differ from our approach.

**Integrating LLMs with Tabular Data** The integration of LLMs with tabular data has led to various applications that extend beyond merely enhancing tabular deep learning. For instance, Yin et al. (2020) pre-trained language models jointly on textual and tabular data, resulting in improved semantic parsing performance. Besides, Han et al. (2022) utilized language model pre-training to better comprehend numerical data, leading to enhancements in question answering and fact verification tasks involving tabular data. In addition, other notable studies include conversing with tables or relational databases (Zha et al., 2023; Hu et al., 2023) and a vision paper on building foundation models for relational databases (Vogel et al., 2023). While these studies focus on improving LLMs' ability to identify answers within tabular data, our paper emphasizes learning on tabular data. This topic involves a data-driven learning procedure applied to tabular data, further expanding the potential benefits of integrating LLMs with structured tables.

**Augmented LLMs** Since the groundbreaking success of LLMs (Brown et al., 2020), ongoing efforts have been made to augment LLMs with capabilities that are difficult to acquire through next-token predictions on pure text alone (Mialon et al., 2023). These efforts can generally be divided into two categories. The first category leverages external sources or tools to gather additional information, thereby endowing LLMs with unprecedented capabilities. Notable examples in this category include ReAct (Yao et al., 2023), which augments LLMs with a simple Wikipedia API, PAL (Gao et al., 2023), which combines LLMs with Python interpreters, and Toolformer (Schick et al., 2023), which teaches LLMs to use multiple tools. The second category introduces an additional tuning pro-

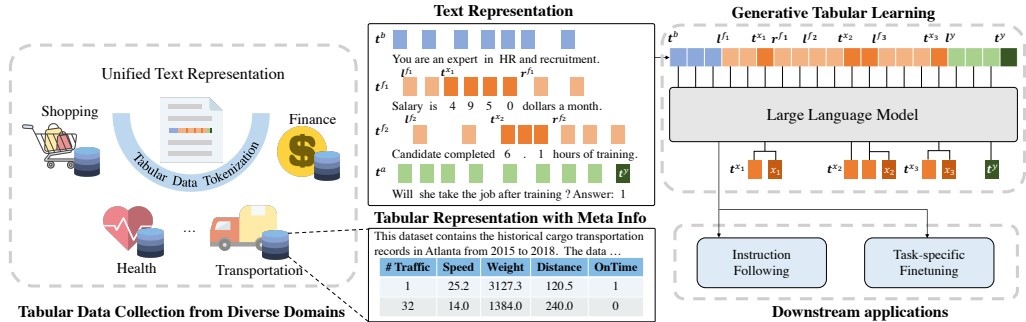

Figure 1: An overview of generative tabuar learning for LLMs.

cedure on new data. For instance, Ouyang et al. (2022) further trained LLMs to align with human values, and Chen et al. (2021) trained LLMs on code data, facilitating remarkable code understanding and generation. From the perspective of building augmented LLMs, this paper also falls into the second category. Our unique contributions include introducing tabular data as a new learning resource for LLMs and developing effective learning objectives tailored to this type of data.

## 3 METHODOLOGY

In this section, we elaborate on our methodology, which starts from a basic problem formulation for traditional tabular learning. Then we introduce our text representation for tabular data, which includes essential meta information, incorporates necessary background, and is well suited as the input of LLMs. Based on this representation, we formally define generative tabular learning for LLMs to acquire foundational knowledge and universal capabilities from diverse tabular datasets. Finally, we discuss several critical aspects of data construction in practice.

**Problem Formulation.** Considering a tabular task $\mathcal{T} : \mathbb{R}^M \to \mathbb{R}^1$, mapping a tabular instance $\boldsymbol{x} \in \mathbb{R}^M$ with $M$ features ($\{x_i\}_{i=1}^M$) to a prediction target $y \in \mathbb{R}^1$, typical solutions for learning on tabular data focus on building a discriminative model to learn the dependency between the target and features, $p(y|\boldsymbol{x})$, from training data. Different tabular learning tasks usually have dramatic differences in feature spaces, including different feature dimensions, different portions of numerical and categorical features, as well as feature values in various scales, and also require different kinds of outputs to support classification or regression. Such significant heterogeneity across tasks poses significant challenges to cross-domain tabular learning.

**Unified Text Representation for Tabular Data.** Representing tabular data in pure text leverages the universality of language to address the challenge of expressing heterogeneous data. Besides, using a text representation enables the easy integration of crucial meta information that can hardly be utilized by traditional studies for learning on tabular data, such as feature meanings and background knowledge. Our text representation consists of three parts: 1) the first part specifies necessary background and task descriptions, and optionally includes some examples as in-context demonstrations; 2) the second part describes feature meanings and values of the current instance to be inferred; 3) the third part includes the prediction target as the answer. Formally, we represent the first part as a sequence of text tokens, $\boldsymbol{t}^b = [t_1^b, \cdots, t_{|\boldsymbol{t}^b|}^b]$, where $|\cdot|$ is the operation to denote the length of a token sequence. Besides, we denote the $i$-th feature as $\boldsymbol{t}^{f_i} = [\boldsymbol{l}^{f_i}, \boldsymbol{t}^{x_i}, \boldsymbol{r}^{f_i}]$, where $\boldsymbol{l}^{f_i} = [l_1^{f_i}, \cdots, l_{|\boldsymbol{l}^{f_i}|}^{f_i}]$ includes feature descriptions on the left of feature values, $\boldsymbol{t}^{x_i} = [t_1^{x_i}, \cdots, t_{|\boldsymbol{t}^{x_i}|}^{x_i}]$ denotes the text sequence for the feature value $x_i$, and $\boldsymbol{r}^{f_i} = [r_1^{f_i}, \cdots, r_{|\boldsymbol{r}^{f_i}|}^{f_i}]$ includes remaining feature descriptions on the right of feature values. By concatenating token sequences for all features, we have the overall feature sequence as $\boldsymbol{t}^f = [\boldsymbol{t}^{f_1}, \cdots, \boldsymbol{t}^{f_M}]$. Last, we represent the task answer as $\boldsymbol{t}^a = [\boldsymbol{l}^a, \boldsymbol{t}^y]$, where $\boldsymbol{l}^a = [l_1^a, \cdots, l_{|\boldsymbol{l}^a|}^a]$ denotes the answer prompt before the answer tokens $\boldsymbol{t}^y = [t_1^y, \cdots, t_{|\boldsymbol{t}^y|}^y]$. In this

way, we can express a tabular instance $\boldsymbol{x}$ and the associated target $y$ as a sequence of tokens:

$$[\boldsymbol{t}^b, \boldsymbol{t}^f, \boldsymbol{t}^a] = [\boldsymbol{t}^b, \boldsymbol{l}^{f_1}, \boldsymbol{t}^{x_1}, \boldsymbol{r}^{f_1}, \cdots, \boldsymbol{l}^{f_M}, \boldsymbol{t}^{x_M}, \boldsymbol{r}^{f_M}, \boldsymbol{l}^a, \boldsymbol{t}^y], \qquad (1)$$

which systematically unifies task background ($\boldsymbol{t}^b$), feature meanings ($\{\boldsymbol{l}^{f_i}, \boldsymbol{r}^{f_i}\}_{i=1}^M$), and feature values ($\{\boldsymbol{t}^{x_i}\}_{i=1}^M$) and supports various prediction targets via a variable-length sequence ($\boldsymbol{t}^y$).

**Generative Tabular Learning (GTL).**  Based on the unified text representation for tabular data, we devise generative tabular learning as characterizing the following joint distribution:

$$p(\boldsymbol{x}, y) = p(\boldsymbol{t}^{\boldsymbol{x}}, \boldsymbol{t}^y | \boldsymbol{t}^m) = p(\boldsymbol{t}^y | \boldsymbol{t}^{\boldsymbol{x}}, \boldsymbol{t}^m) \prod_{i=1}^M p(\boldsymbol{t}^{x_i} | \boldsymbol{t}^{<x_i}), \qquad (2)$$

where we introduce additional notations to ensure the concise formulation, using $\boldsymbol{t}^m = [\boldsymbol{t}^b, \boldsymbol{l}^{f_1}, \boldsymbol{r}^{f_1}, \cdots, \boldsymbol{l}^{f_M}, \boldsymbol{r}^{f_M}, \boldsymbol{l}^a]$ to denote all meta information, $\boldsymbol{t}^{\boldsymbol{x}} = [\boldsymbol{t}^{x_1}, \cdots, \boldsymbol{t}^{x_M}]$ to represent all tokens related to feature values, and $\boldsymbol{t}^{<x_i}$ to include all tokens ahead of $\boldsymbol{t}^{x_i}$ in equation 1. Here $p(\boldsymbol{x}, y)$ denotes the joint distribution in the initial feature and label spaces, while $p(\boldsymbol{t}^{\boldsymbol{x}}, \boldsymbol{t}^y | \boldsymbol{t}^m)$ represents the same joint distribution conditioned on all meta information using the text represen-tation, which can further be decoupled autoregressively into $p(\boldsymbol{t}^y | \boldsymbol{t}^{\boldsymbol{x}}, \boldsymbol{t}^m) \prod_{i=1}^M p(\boldsymbol{t}^{x_i} | \boldsymbol{t}^{<x_i})$. It is thus straightforward to leverage LLMs, especially those using auto-regressive architectures, to char-acterize this decoupled formulation. The only modification needed is to mask out the losses on meta-information tokens. In addition, to facilitate the understanding of numerical information, we encourage LLMs to recover the numerical information from its text representation, namely capturing $p(x_i | \boldsymbol{t}^{x_i})$, which can be easily implemented as adding an extra prediction head during training and just overlooking this head in inference. In summary, our generative tabular learning is employing LLMs to characterize the following joint distribution:

$$p(\boldsymbol{t}^y | \boldsymbol{t}^{\boldsymbol{x}}, \boldsymbol{t}^m) \prod_{i=1}^M p(\boldsymbol{t}^{x_i} | \boldsymbol{t}^{<x_i}) p(x_i | \boldsymbol{t}^{x_i}), \quad (\boldsymbol{x}, y) \in \mathcal{T}, \quad \mathcal{T} \in \mathcal{D}^{\mathcal{T}}, \qquad (3)$$

where $\mathcal{D}^{\mathcal{T}}$ denotes the family of tabular tasks. In this way, we turn LLMs into TabFMs that have acquired foundational knowledge and universal capabilities from diverse tabular datasets across dif-ferent domains. After the stage of GTL, TabFMs can be used in either instruction-following or fine-tuning scenarios for new tabular tasks.

**Data Construction in Practice.**  Here we would like briefly recap critical aspects of data con-truction in practice to support the modeling of equation 3. The first is to express essential meta information, such as task background and feature meanings, precisely and concisely. A precise ex-pression helps to convey complete semantic information to facilitate the generalization of LLMs, while a concise statement ensures the resulting sequence is not too long. Second, we find that adding additional data examples as demonstrations into $\boldsymbol{t}^b$ also helps to boost the in-context gen-eralization (Brown et al., 2020). Thus, we adopt a hybrid approach, combining the data using no demonstration with the one having in-context examples as demonstrations. Besides, more details on our data construction can be found in Appendix A.2.

## 4  EXPERIMENTS

We carry out comprehensive experiments to showcase the effectiveness of our proposed GTL in constructing TabFMs. Our evaluation primarily encompasses two usage scenarios for downstream tasks. The first, detailed in Section 4.1, assesses the capability of instruction following in new tabular tasks without requiring any additional tuning. The second, depicted in Section 4.2, evaluates the performance of fine-tuning with varying amounts of training data.

**Implementation of GTL.**  Before exploring the two experimental sections, we first introduce the implementation of GTL in this work. Initially, we gather hundreds of tabular learning datasets span-ning diverse domains from Kaggle[2]. After a data selection process that prioritizes diverse domains

---

[2]https://www.kaggle.com/datasets

Table 1: Performance comparisons on instruction following for different tasks with varying numbers of contexts. We show average AUROC scores with the standard deviation denoted as subscript. The best scores are highlighted in bold, while the second-best scores are underlined.

| #Contexts | Method | Bank | Blood | C. Hous. | Car | Creditg | Diabetes | Heart | Income | Jungle |
|---|---|---|---|---|---|---|---|---|---|---|
| 0 | GPT-3.5 | $0.51_{00}$ | $0.56_{00}$ | $0.50_{00}$ | - | $0.48_{00}$ | $0.69_{00}$ | $0.62_{00}$ | $0.64_{00}$ | $0.49_{00}$ |
| | GPT-4 | $\underline{0.79}_{00}$ | $\underline{0.71}_{00}$ | $\mathbf{0.81}_{00}$ | $\underline{0.73}_{00}$ | $\underline{0.58}_{00}$ | $\mathbf{0.83}_{00}$ | $\mathbf{0.86}_{00}$ | $\underline{0.83}_{00}$ | $\mathbf{0.61}_{00}$ |
| | T0 (TabLLM) | $0.63_{01}$ | $0.61_{04}$ | $\underline{0.61}_{01}$ | $\mathbf{0.82}_{02}$ | $0.53_{05}$ | $0.68_{06}$ | $0.54_{04}$ | $\mathbf{0.84}_{00}$ | $0.60_{00}$ |
| | LLaMA | $0.57_{01}$ | $0.56_{02}$ | $0.52_{01}$ | $0.66_{02}$ | $0.56_{06}$ | $0.66_{05}$ | $0.63_{03}$ | $0.64_{00}$ | $0.58_{00}$ |
| | LLaMA-GTL | $\mathbf{0.85}_{00}$ | $\mathbf{0.72}_{03}$ | $0.49_{01}$ | $0.71_{01}$ | $\mathbf{0.60}_{06}$ | $\underline{0.82}_{03}$ | $\underline{0.65}_{04}$ | $0.78_{01}$ | $0.59_{00}$ |
| 4 | GPT-3.5 | $0.58_{00}$ | $0.53_{00}$ | $0.58_{00}$ | - | $0.50_{00}$ | $0.70_{00}$ | $0.68_{00}$ | $0.68_{00}$ | $0.54_{00}$ |
| | GPT-4 | $\underline{0.76}_{00}$ | $\mathbf{0.72}_{00}$ | $\mathbf{0.79}_{00}$ | $\mathbf{0.80}_{00}$ | $\mathbf{0.60}_{00}$ | $\underline{0.78}_{00}$ | $0.66_{00}$ | $\mathbf{0.80}_{00}$ | $\underline{0.63}_{00}$ |
| | TabPFN | $0.59_{14}$ | $0.52_{08}$ | $0.63_{13}$ | $0.64_{06}$ | $0.58_{08}$ | $0.61_{13}$ | $\mathbf{0.84}_{06}$ | $0.73_{08}$ | $\mathbf{0.65}_{08}$ |
| | T0 (TabLLM) | $0.49_{01}$ | $0.57_{03}$ | $0.50_{04}$ | $0.52_{05}$ | $0.42_{03}$ | $0.49_{00}$ | $0.50_{04}$ | $0.53_{00}$ | $0.48_{01}$ |
| | LLaMA | $0.55_{01}$ | $0.58_{02}$ | $0.65_{01}$ | $\underline{0.77}_{02}$ | $0.50_{03}$ | $0.65_{04}$ | $0.66_{03}$ | $\underline{0.75}_{00}$ | $0.53_{01}$ |
| | LLaMA-GTL | $\mathbf{0.85}_{00}$ | $\underline{0.68}_{05}$ | $\underline{0.72}_{01}$ | $\mathbf{0.80}_{01}$ | $\underline{0.59}_{04}$ | $\mathbf{0.81}_{04}$ | $\underline{0.71}_{03}$ | $\mathbf{0.80}_{00}$ | $0.56_{00}$ |
| 8 | GPT-3.5 | $0.62_{00}$ | $0.56_{00}$ | $0.54_{00}$ | - | $0.48_{00}$ | $0.67_{00}$ | $0.65_{00}$ | $0.70_{00}$ | $0.53_{00}$ |
| | GPT-4 | $\underline{0.81}_{00}$ | $\underline{0.64}_{00}$ | $\mathbf{0.79}_{00}$ | $\mathbf{0.82}_{00}$ | $\underline{0.59}_{00}$ | $\underline{0.76}_{00}$ | $0.69_{00}$ | $\underline{0.80}_{00}$ | $\underline{0.63}_{00}$ |
| | TabPFN | $0.66_{08}$ | $\underline{0.64}_{04}$ | $0.63_{11}$ | $0.75_{05}$ | $\mathbf{0.59}_{03}$ | $0.67_{11}$ | $\mathbf{0.88}_{05}$ | $0.71_{09}$ | $\mathbf{0.72}_{04}$ |
| | T0 (TabLLM) | $0.48_{00}$ | $0.54_{04}$ | $0.52_{08}$ | $0.57_{05}$ | $0.47_{04}$ | $0.48_{00}$ | $0.51_{03}$ | $0.51_{00}$ | $0.50_{01}$ |
| | LLaMA | $0.56_{01}$ | $0.63_{02}$ | $0.69_{00}$ | $0.80_{01}$ | $0.52_{05}$ | $0.67_{04}$ | $\underline{0.74}_{03}$ | $0.78_{01}$ | $0.55_{00}$ |
| | LLaMA-GTL | $\mathbf{0.85}_{00}$ | $\mathbf{0.68}_{02}$ | $\underline{0.75}_{01}$ | $\underline{0.81}_{02}$ | $0.56_{06}$ | $\mathbf{0.83}_{04}$ | $\underline{0.74}_{02}$ | $\mathbf{0.81}_{01}$ | $0.57_{00}$ |

and maintains a reasonable range of instances and features, we are left with 115 tabular datasets. To balance the number of instances across different datasets, we randomly sample up to $2,048$ instances from each tabular dataset for GTL. Subsequently, we convert these tabular datasets into our unified text representations, as described in Section 3. More details about data collection and prompt generation can be found in A.2 and A.3. Concerning the learning model, we utilize GPT-4 (OpenAI, 2023) to generate descriptions about task background and feature meanings, and we also involve manual verifications to adjust and correct some descriptions. We employ LLaMA-2-7B (Touvron et al., 2023) as our base LLM, which will be simply referred to as 'LLaMA' in the following sections for brevity. Additionally, we denote the obtained TabFM after the GTL stage as LLaMA-GTL. More implementation details about LLaMA-GTL can be found in Appendix B.2.

**Datasets and Metrics for Evaluation.** To maintain consistency with previous studies, we adopt the nine datasets constructed by Hegselmann et al. (2023) for evaluation, spanning diverse domains such as healthcare, finance, economics, and gaming. And each dataset contains five random splits to produce robust evaluations. A self-contained introduction to these datasets can be found in Appendix A.1. Notably, our constructed datasets for GTL are non-overlapping with these evaluation datasets, ensuring no data leakage issues. However, two evaluation datasets, Bank and Diabetes, contain similar knowledge to some of the GTL datasets. Applying LLaMA-GTL to these two datasets can be considered a form of in-domain transfer, while adapting to the other seven evaluation datasets represents some degree of out-of-domain transfer. A thorough discussion on the connections between evaluation datasets and our 115 GTL datasets is included in Appendix A.4. Since we focus on classification tasks in this work, following Hegselmann et al. (2023), we use the area under the receiver operating characteristic (AUROC) score as the primary evaluation metric.

### 4.1 INSTRUCTION FOLLOWING

**Experimental Setups.** In this section, we focus on evaluating the capability of instruction following in new tabular tasks, enforcing no additional tuning. The specific scenarios include zero-shot and in-context inferences with varying number of context examples. We include three types of baselines in our experiments. The first type, consisting of GPT-3.5 (Brown et al., 2020) and GPT-4 (OpenAI, 2023), represents closed-source black-box LLMs. The second type, including T0 (TabLLM) (Hegselmann et al., 2023) and LLaMA (Touvron et al., 2023), denotes open-source LLMs that we can fully control. Note that we leverage the LLaMA-2-7B version as the base LLM for GTL,

and we name the resulting TabFM as LLaMA-GTL. The third type comprises a special prior-data fitted network, TabPFN (Hollmann et al., 2023), which possesses in-context capabilities for new tabular tasks but cannot support zero-shot inference. More implementation details about these baselines can be found in Appendix B.1. Table1 presents the overall performance comparisons on instruction following between our model, LLaMA-GTL, and other baselines. In the following sections, we will elaborate on our findings and insights in interpreting these results.

**Zero-shot vs. In-context.** Before delving into the details of comparing different models, a general yet notable observation is that *introducing in-context examples does not necessarily lead to performance improvements over zero-shot inference in new tabular tasks*. This observation applies to all baselines, including the powerful GPT-4, and our model, LLaMA-GTL. A possible explanation for this observation could be the conflict between directly transferring the general knowledge held by the model and quickly gaining statistical knowledge from very few in-context examples. For certain datasets that share more common knowledge with the pretrained model, zero-shot performance could significantly surpass in-context performance when very few in-context examples do not provide robust statistical knowledge, thereby acting as an interfering effect. Conversely, for datasets with specialized domain-specific knowledge, in-context examples may significantly improve zero-shot inference if the model can quickly identify effective patterns from these examples. Since different models may cover dramatically different general knowledge, given the diversified corpora they have been trained on, the performance variations between zero-shot and in-context inferences are also divergent for these models.

**LLaMA-GTL significantly improves LLaMA in most zero-shot scenarios.** The zero-shot performance of the base LLaMA model is suboptimal compared to other methods, indicating that LLaMA is not well-equipped with sufficient knowledge for tabular prediction tasks. With the generative tabular learning, LLaMA-GTL significantly improves the predictive performance of LLaMA on most datasets, demonstrating the promising capability of knowledge transfer from the pretuning data to downstream tasks. Specifically, there exist similar datasets for Diabetes and Bank in the GTL corpora, which contain similar domain knowledge but different data schemas and prediction targets. In such in-domain knowledge transfer cases, LLaMA-GTL successfully leverages the knowledge acquired during the pretuning stage to achieve substantial performance gains, showcasing the great potential of GTL to enhance downstream tasks by training on more diverse datasets. For other evaluation datasets, we do not find similar datasets in the GTL corpora that explicitly carry similar information. Nevertheless, even under such out-of-domain knowledge transfer cases, we observe significant improvements in Blood, Car, and Income datasets compared with the base LLaMA model. The knowledge required to make accurate predictions on these datasets (forecasting the willingness to donate blood based on previous donations, car prices based on their conditions, and income based on social status and identity) can be transferred from general training materials without explicit domain overlap. However, there are failure cases for LLaMA-GTL, including Heart and C. Hous. datasets. The base LLaMA does not significantly benefit from GTL on these two datasets, as the GTL corpora do not include any relevant information helpful to these two tasks. Additionally, almost all LLMs struggle on Creditg and Jungle datasets due to the unclear predictable patterns without the bank's credit-giving strategy and the complex rules of Jungle chess.

**LLaMA-GTL also improves LLaMA in in-context scenarios.** With additional in-context examples, LLaMA-GTL continues to improve the prediction performance of LLaMA on Car, Diabetes, and Income datasets. Compared to the zero-shot performance, LLaMA-GTL has demonstrated its capabilities in learning from very few in-context examples and significantly boosting the performance on C. Hours. and Heart datasets. Additionally, we observe that the base LLaMA itself has also gained some benefits from in-context examples. However, there are also some failure cases, in which the in-context performance is inferior to the zero-shot performance. This indicates a significant challenge for all models in achieving a better balance between the general knowledge learned and the data-driven knowledge derived from a very limited number of in-context examples.

**Comparing LLaMA-GTL with other baselines.** Providing a fair comparison of all the baselines is challenging, as they typically employ different pre-training corpora and have varying model scales. Instead of arguing which one is the best solution, we compare their performance differences and attempt to gain insights that may guide future research. First, we find that similar to

Table 2: Performance comparisons of fine-tuning new tasks. The methods are split in groups representing linear and tree models, deep tabular models, and LLM-based models. The numbers in the table represent the average AUROC scores with the standard deviation in the subscript. The best scores achieved are highlighted in bold and the second best scores are underlined.

| #Shots | Method | Bank | Blood | C. Hous. | Car | Creditg | Diabetes | Heart | Income | Jungle |
|---|---|---|---|---|---|---|---|---|---|---|
| | Logistic Reg. | $0.84_{.02}$ | $\mathbf{0.74_{.02}}$ | $0.88_{.01}$ | $0.93_{.02}$ | $0.66_{.07}$ | $0.80_{.02}$ | $0.91_{.01}$ | $0.83_{.03}$ | $0.79_{.01}$ |
| | LightGBM | $0.77_{.03}$ | $0.69_{.04}$ | $0.81_{.02}$ | $0.85_{.06}$ | $0.61_{.09}$ | $0.79_{.02}$ | $0.91_{.01}$ | $0.78_{.03}$ | $0.79_{.02}$ |
| | XGBoost | $0.83_{.02}$ | $0.68_{.05}$ | $0.82_{.04}$ | $0.91_{.02}$ | $0.67_{.06}$ | $0.73_{.05}$ | $0.91_{.01}$ | $0.82_{.02}$ | $\mathbf{0.81_{.02}}$ |
| | SAINT | $0.81_{.03}$ | $0.67_{.05}$ | $0.81_{.02}$ | $0.92_{.02}$ | $0.66_{.06}$ | $0.79_{.03}$ | $0.90_{.04}$ | $\mathbf{0.84_{.02}}$ | $\mathbf{0.81_{.01}}$ |
| 64 | TabNet | $0.71_{.06}$ | $0.63_{.06}$ | $0.72_{.03}$ | $0.73_{.07}$ | $0.56_{.05}$ | $0.71_{.04}$ | $0.83_{.05}$ | $0.71_{.04}$ | $0.73_{.04}$ |
| | NODE | $0.78_{.02}$ | $0.71_{.05}$ | $0.80_{.01}$ | $0.80_{.02}$ | $0.63_{.04}$ | $0.77_{.04}$ | $0.88_{.02}$ | $0.75_{.02}$ | $0.75_{.04}$ |
| | TabPFN | $0.82_{.03}$ | $0.73_{.04}$ | $\mathbf{0.89_{.01}}$ | $\mathbf{0.97_{.00}}$ | $\mathbf{0.70_{.07}}$ | $0.82_{.03}$ | $\mathbf{0.92_{.02}}$ | $0.82_{.04}$ | $\mathbf{0.81_{.01}}$ |
| | TabLLM | $0.69_{.03}$ | $0.68_{.04}$ | $0.77_{.04}$ | $0.96_{.02}$ | $\mathbf{0.70_{.07}}$ | $0.73_{.03}$ | $0.91_{.01}$ | $\mathbf{0.84_{.02}}$ | $0.78_{.02}$ |
| | LLaMA | $0.62_{.02}$ | $0.66_{.03}$ | $0.57_{.04}$ | $0.90_{.02}$ | $0.67_{.09}$ | $0.78_{.05}$ | $0.88_{.02}$ | $\mathbf{0.84_{.02}}$ | $0.63_{.04}$ |
| | LLaMA-GTL | $\mathbf{0.86_{.01}}$ | $0.72_{.05}$ | $0.78_{.04}$ | $0.96_{.01}$ | $0.70_{.09}$ | $\mathbf{0.83_{.04}}$ | $0.88_{.05}$ | $\mathbf{0.84_{.01}}$ | $0.69_{.04}$ |
| | Logistic Reg. | $0.89_{.00}$ | $\mathbf{0.76_{.03}}$ | $0.91_{.00}$ | $0.98_{.00}$ | $\mathbf{0.76_{.02}}$ | $0.83_{.02}$ | $\mathbf{0.93_{.01}}$ | $0.88_{.00}$ | $0.80_{.00}$ |
| | LightGBM | $0.89_{.00}$ | $0.67_{.05}$ | $0.92_{.00}$ | $0.99_{.01}$ | $0.75_{.02}$ | $0.79_{.03}$ | $0.92_{.01}$ | $0.88_{.00}$ | $\mathbf{0.91_{.00}}$ |
| | XGBoost | $\mathbf{0.90_{.01}}$ | $0.67_{.06}$ | $0.92_{.01}$ | $0.99_{.01}$ | $0.75_{.03}$ | $0.80_{.01}$ | $0.92_{.01}$ | $0.88_{.00}$ | $\mathbf{0.91_{.01}}$ |
| | SAINT | $0.88_{.01}$ | $0.73_{.02}$ | $0.91_{.02}$ | $0.99_{.00}$ | $0.73_{.03}$ | $0.77_{.03}$ | $0.92_{.01}$ | $0.88_{.00}$ | $0.90_{.00}$ |
| 512 | TabNet | $0.83_{.03}$ | $0.72_{.02}$ | $0.87_{.01}$ | $0.98_{.01}$ | $0.66_{.04}$ | $0.74_{.07}$ | $0.88_{.03}$ | $0.83_{.02}$ | $0.84_{.01}$ |
| | NODE | $0.86_{.01}$ | $\mathbf{0.76_{.03}}$ | $0.87_{.01}$ | $0.96_{.01}$ | $0.70_{.02}$ | $0.83_{.02}$ | $0.92_{.03}$ | $0.83_{.01}$ | $0.80_{.00}$ |
| | TabPFN | $\mathbf{0.90_{.00}}$ | $\mathbf{0.76_{.03}}$ | $\mathbf{0.93_{.00}}$ | $\mathbf{1.00_{.00}}$ | $0.75_{.02}$ | $0.81_{.02}$ | $0.92_{.02}$ | $0.87_{.01}$ | $\mathbf{0.91_{.00}}$ |
| | TabLLM | $0.88_{.01}$ | $0.68_{.04}$ | $0.86_{.02}$ | $\mathbf{1.00_{.00}}$ | $0.72_{.02}$ | $0.78_{.04}$ | $0.92_{.01}$ | $\mathbf{0.89_{.01}}$ | $0.89_{.01}$ |
| | LLaMA | $0.77_{.02}$ | $0.72_{.05}$ | $0.86_{.02}$ | $0.99_{.00}$ | $0.72_{.04}$ | $0.83_{.04}$ | $0.92_{.02}$ | $\mathbf{0.89_{.01}}$ | $0.85_{.03}$ |
| | LLaMA-GTL | $\mathbf{0.90_{.00}}$ | $0.75_{.04}$ | $0.89_{.02}$ | $0.99_{.01}$ | $0.74_{.05}$ | $\mathbf{0.85_{.03}}$ | $\mathbf{0.93_{.02}}$ | $\mathbf{0.89_{.01}}$ | $0.89_{.01}$ |
| | Logistic Reg. | $0.91_{.00}$ | $\mathbf{0.76_{.03}}$ | $0.92_{.00}$ | $0.98_{.00}$ | $\mathbf{0.79_{.03}}$ | $0.83_{.02}$ | $\mathbf{0.93_{.01}}$ | $0.90_{.00}$ | $0.81_{.00}$ |
| | LightGBM | $\mathbf{0.94_{.00}}$ | $0.74_{.04}$ | $\mathbf{0.97_{.00}}$ | $\mathbf{1.00_{.00}}$ | $0.78_{.02}$ | $0.83_{.03}$ | $\mathbf{0.94_{.01}}$ | $\mathbf{0.93_{.00}}$ | $0.98_{.00}$ |
| | XGBoost | $\mathbf{0.94_{.00}}$ | $0.71_{.04}$ | $\mathbf{0.97_{.00}}$ | $\mathbf{1.00_{.00}}$ | $0.78_{.04}$ | $0.84_{.03}$ | $\mathbf{0.94_{.01}}$ | $\mathbf{0.93_{.00}}$ | $0.98_{.00}$ |
| | SAINT | $0.93_{.00}$ | $0.74_{.03}$ | $0.95_{.00}$ | $\mathbf{1.00_{.00}}$ | $0.77_{.04}$ | $0.83_{.03}$ | $\mathbf{0.93_{.01}}$ | $0.91_{.00}$ | $\mathbf{1.00_{.00}}$ |
| All | TabNet | $0.93_{.00}$ | $0.71_{.03}$ | $0.96_{.00}$ | $\mathbf{1.00_{.00}}$ | $0.64_{.03}$ | $0.81_{.03}$ | $0.89_{.03}$ | $0.92_{.00}$ | $0.99_{.00}$ |
| | NODE | $0.76_{.02}$ | $0.74_{.03}$ | $0.87_{.01}$ | $0.93_{.01}$ | $0.65_{.03}$ | $0.83_{.03}$ | $0.92_{.03}$ | $0.82_{.00}$ | $0.81_{.00}$ |
| | TabPFN | $0.91_{.00}$ | $0.74_{.03}$ | $0.94_{.00}$ | $\mathbf{1.00_{.00}}$ | $0.75_{.03}$ | $0.81_{.03}$ | $0.92_{.02}$ | $0.89_{.00}$ | $0.93_{.00}$ |
| | TabLLM | $0.92_{.00}$ | $0.70_{.04}$ | $0.95_{.00}$ | $\mathbf{1.00_{.00}}$ | $0.70_{.02}$ | $0.80_{.04}$ | $\mathbf{0.94_{.01}}$ | $0.92_{.00}$ | $\mathbf{1.00_{.00}}$ |
| | LLaMA | $\mathbf{0.94_{.00}}$ | $0.72_{.04}$ | $\mathbf{0.97_{.00}}$ | $\mathbf{1.00_{.00}}$ | $0.76_{.07}$ | $0.84_{.03}$ | $0.93_{.01}$ | $\mathbf{0.93_{.00}}$ | $\mathbf{1.00_{.00}}$ |
| | LLaMA-GTL | $\mathbf{0.94_{.00}}$ | $0.75_{.05}$ | $0.96_{.00}$ | $\mathbf{1.00_{.00}}$ | $0.76_{.06}$ | $\mathbf{0.85_{.04}}$ | $0.93_{.01}$ | $\mathbf{0.93_{.00}}$ | $\mathbf{1.00_{.00}}$ |

LLaMA, GPT-3.5 does not excel in these tabular prediction tasks. In contrast, GPT-4 exhibits remarkable zero-shot and in-context capabilities in most cases, demonstrating comprehensive and rich domain knowledge learned from its pre-training process. However, it is difficult to verify whether similar information has already been fed into GPT-4, as it is a black-box model, and all these evaluation datasets are publicly available. Nonetheless, LLaMA-GTL, with only 7B parameters, approaches GPT-4's performance in many cases and even surpasses it on Bank, Blood, and Creditg datasets. These observations confirm that GTL helps LLaMA successfully acquire foundational knowledge and universal capabilities across diverse tabular datasets. Additionally, we note that the T0 (TabLLM) model occasionally demonstrates surprising performance, such as outperforming GPT-4 on the Car and Income datasets. We conjecture the underlying reason may also be related to introducing relevant information into pre-training and the prompt-engineering efforts made by Hegselmann et al. (2023). Another interesting baseline is TabPFN. Although it does not support zero-shot inference, its in-context results in some cases, such as Jungle and Heart, are very impressive, outperforming all LLM counterparts. This observation inspires us to combine the strengths of both prior-data fitted networks and LLMs to create more advanced tabular models.

## 4.2 Task-specific Finetuning

**Experimental setups.** We adopt the same evaluation datasets to further study fine-tuning performance in new tasks. We follow the data partitioning and hyperparameter selection process outlined in Hegselmann et al. (2023). For each dataset, we select 64 shots, 512 shots, and all training samples to study fine-tuning under different data scales. We omit the fewer shot settings ($\leq 32$) because

1) the resulting performance exhibits large variances, and 2) we may resort to in-context learning with limited training examples, given the potential for much larger context lengths in the future. We compare our approach with three types of methods, including linear and tree models, deep neural networks, and LLM-based solutions. Tree models include XGBoost (Chen & Guestrin, 2016) and LightGBM (Ke et al., 2017). Deep neural networks comprise SAINT (Somepalli et al., 2021), Tab-Net (Arik & Pfister, 2021), NODE (Shwartz-Ziv & Armon, 2022a), and TabPFN (Hollmann et al., 2023). LLM-based baselines include TabLLM (Hegselmann et al., 2023) and LLaMA (Touvron et al., 2023). More implementation details can be found in Section B.

**LLaMA-GTL significantly improves the fine-tuning performance of LLaMA in few-shot scenarios, while maintaining similar performance given ample training data.** First, we observe that models specifically designed for tabular data modeling exhibit strong predictive capabilities when provided with examples for tuning. In contrast, LLaMA struggles to capture useful patterns from few-shot samples of Bank, Blood, C. Hous., and Jungle datasets, indicating that it lacks the capability for tabular prediction and cannot quickly adapt to new domains. When employing LLaMA-GTL, the performance on these datasets significantly improves, matching and even surpassing state-of-the-art models on most datasets. These observations suggest that the foundational knowledge learned through GTL also aids transfer in fine-tuning scenarios, demonstrating extreme data efficiency and rapid generalization. Additionally, we note that given ample training data, GTL has negligible effects on the fine-tuning, indicating that sufficient training data provides effective statistical patterns that dominate generalization performance.

Although LLaMA-GTL matches and even surpasses the state-of-the-art solutions for learning on tabular data on most datasets, we also observe some failure cases. For example, as for learning with 64 shots on the Jungle dataset, LLaMA-GTL obtains the average AUROC score of 0.69, though improving the average score of LLaMA (0.63), which is far behind the results, around 0.80, of many tabular models. We conjecture the underlying reasons may be related to the text representation used, involving different prompts and tokenization schemes, or the general knowledge hold by LLMs, which is related to the pre-training corpora used. Specifically, Jungle is a special dataset about gaming, which neither LLaMA nor our GTL data has relevant information about.

## 5 DISCUSSION AND CONCLUSION

In this paper, we proposed Tabular Foundation Models (TabFMs) to address the limitations of existing transferable tabular models by supporting both fine-tuning and instruction following for new tasks and acquiring foundational knowledge and universal capabilities across diverse domains. We have developed GTL to transform LLMs into TabFMs and conducted extensive evaluations to showcase the effectiveness of our approach.

**Limitations.** Despite the promising results, several limitations and challenges remain. First, our current implementation of GTL relies on a limited number of tabular datasets from diverse domains. Expanding the dataset collection could further enhance the capabilities of TabFMs, ensuring a more comprehensive understanding of tabular data. Second, the limited context length of most existing LLMs, including our choice, restricts the number of features that can be considered and the number of in-context examples provided. Developing LLMs with longer context lengths or devising strategies to handle larger contexts efficiently could lead to improved performance in real-world applications with many features. Furthermore, it remains uncertain whether learning numerical information in text representations can be further enhanced. The TabPFN baseline, operating on the numerical representation of tabular data, demonstrates remarkable performance in few-shot scenarios, suggesting potential benefits from combining the strengths of both representations for tabular data to build even more powerful models. Future research should investigate methods to integrate the best of both worlds, leveraging the power of LLMs and specialized numerical models.

Meanwhile, we hope that our critical findings and insights, such as the existence of a gap between the general knowledge held by LLMs and the data-driven knowledge required to understand tabular data, the introduction of GTL significantly enhancing LLMs' performance on tabular tasks, and the potential for tabular data to serve as a new corpus resource for LLMs, will inspire and expedite future research on developing more powerful foundation models for tabular data.

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

# A DATA

## A.1 TABLLM DATASETS

In our experiment, we employ the same baseline datasets used in TabLLM, maintaining the original train, validation, and test splits, as well as adhering to the few-shot sampling procedure. This approach ensures a consistent and reliable comparison between our results and those reported in TabLLM (Hegselmann et al., 2023). A brief description of the baseline datasets utilized in this study is provided below:

- **Bank** (Kadra et al., 2021) comprises information from a direct marketing campaign conducted by a Portuguese banking institution (Moro et al., 2014). The objective is to predict whether a customer subscribed to a term deposit. The dataset includes 45,211 rows with 16 features, of which 5,289 labels are positive.

- **Blood** (Kadra et al., 2021) contains data from a blood transfusion service in Taiwan (Yeh et al., 2009), featuring 4 attributes of 748 donors. The label indicates whether donors returned for another donation, with 178 positive instances.

- **C. Hous.** (Grinsztajn et al., 2022a) consists of eight attributes for 20,640 districts in California, with the goal of predicting the median house value in each district (Pace & Barry, 1997). Following (Grinsztajn et al., 2022a), we created a balanced classification task to predict whether the house value is below or above the median, resulting in 10,317 positive instances.

- **Car** (Kadra et al., 2021) features entries for various cars, characterized by six attributes. The task involves a multiclass classification problem to evaluate each car's state. The dataset comprises 1,728 rows, with the four classes having a distribution of 1210, 384, 65, and 69 examples.

- **Credit-g** (Kadra et al., 2021) describes 1,000 individuals from Germany seeking credit and includes 20 attributes. The label predicts whether they have a good or bad risk, with 700 classified as good.

- **Diabetes** (from Kaggle[3]) was collected by the National Institute of Diabetes and Digestive and Kidney Diseases (Smith et al., 1988). The dataset contains 768 rows, each representing women of Pima Indian heritage with eight clinical variables. The binary classification task determines whether a person has diabetes, with 268 positive cases.

- **Heart** (from Kaggle[4]) includes data from four different hospitals (Detrano et al., 1989). Each row features 11 clinical variables of a patient. The binary classification task predicts coronary artery disease, with 508 out of 918 patients being positive.

- **Income** (Kadra et al., 2021; Borisov et al., 2022), also known as Adult, contains rows for 48,842 individuals with twelve attributes gathered from the 1994 U.S. Census (Kohavi et al., 1996; Dua & Graff, 2017). The task is to predict whether each person has an annual income exceeding $50,000. The dataset has 11,687 positive labels.

- **Jungle** (Kadra et al., 2021) is a collection of 44,819 end game positions of Jungle Chess (van Rijn & Vis, 2014). Each game is described with 6 attributes, and the goal is to predict whether the white player will win (23,062 positive instances).

## A.2 115 TABULAR DATASETS FROM KAGGLE

To enable Large Language Models (LLMs) to acquire foundational knowledge and capabilities across various domains through generative tabular learning, it is essential to compile a large-scale and diverse collection of tabular datasets. We have meticulously curated a selection of 115 public datasets, which consist of high-quality, usable tabular classification datasets sourced from Kaggle. The total number of data samples amounts to 1,102,760. A comprehensive description of each dataset is provided in Table 3. For the majority of the datasets, the tags are annotated in Kaggle, while for others without specific domain tags, they are generated by GPT-4 (OpenAI, 2023).

---

[3] https://www.kaggle.com/datasets/uciml/pima-indians-diabetes-database
[4] https://www.kaggle.com/fedesoriano/heart-failure-prediction

Table 3: Summary of 115 tabular datasets from Kaggle.

| Dataset | Tags | #Rows | #Classes | #Features |
|---|---|---|---|---|
| preterm-data-set | physics, health | 58 | 2 | 5 |
| obesity-classification-dataset | health | 108 | 4 | 5 |
| zoo-animals-extended-dataset | animals | 113 | 7 | 17 |
| top-10-machine-learning-datasets | cancer | 117 | 2 | 14 |
| higher-education-students-performance-evaluation | research, education | 145 | 8 | 30 |
| credit-score-classification-dataset | finance, banking, investing | 164 | 3 | 7 |
| spotify-recommendation | music, programming | 195 | 2 | 13 |
| -world-metro | global, business | 198 | 6 | 7 |
| glass-identification-data-set | materials | 214 | 6 | 9 |
| factors-affecting-campus-placement | education, business | 215 | 2 | 13 |
| world-bank-country-and-lending-groups | economics | 218 | 4 | 6 |
| entrepreneurial-competency-in-university-students | education, business | 219 | 2 | 16 |
| star-dataset | physics, astronomy | 240 | 6 | 6 |
| mugla-region-wildfirejune122-2022 | environment, safety, geospatial | 267 | 2 | 9 |
| human-genetic-data | biology | 294 | 8 | 10 |
| hr-competency-scores-for-screening | education, employment | 300 | 2 | 9 |
| biomechanical-features-of-orthopedic-patients | health | 310 | 2 | 5 |
| vertebralcolumndataset | biology, medicine | 310 | 2 | 6 |
| crystal-system-properties-for-liion-batteries | business | 339 | 3 | 9 |
| disease-symptoms-and-patient-profile-dataset | medicine | 349 | 2 | 9 |
| dementia-prediction-dataset | medicine | 373 | 4 | 12 |
| predict-diabetes-based-on-diagnostic-measures | diabetes | 390 | 2 | 14 |
| migraine-classification | health | 400 | 7 | 23 |
| cirrhosis-prediction-dataset | health | 418 | 3 | 18 |
| birds-bones-and-living-habits | biology | 420 | 6 | 10 |
| yugioh-normal-monster-cards | games | 478 | 8 | 5 |
| employee-satisfaction-index-dataset | employment, business, software | 500 | 2 | 11 |
| loan-application-data | banking | 500 | 2 | 12 |
| bmidataset | health | 500 | 6 | 3 |
| early-diabetes-classification | diabetes, health | 520 | 2 | 16 |
| urinary-biomarkers-for-pancreatic-cancer | biology, cancer, medicine | 590 | 3 | 12 |
| fairs-extramarital-affairs-data | psychology, people | 601 | 6 | 8 |
| home-loan-approval | business | 614 | 2 | 11 |
| hepatitis-c-dataset | cancer, health | 615 | 5 | 12 |
| breast-cancer-wisconsin-benign-or-malignant | cancer, health | 683 | 2 | 9 |
| breastcancerwisconsin | cancer | 699 | 2 | 9 |
| pokmon-legendary-data | games, entertainment | 801 | 2 | 12 |
| titanic-dataset | history, transportation | 891 | 2 | 10 |
| unicorn-decacorn-hectocron-in-2021 | finance, business | 936 | 3 | 7 |
| tour-travels-customer-churn-prediction | business | 954 | 2 | 6 |
| passenger-list-for-the-estonia-ferry-disaster | health | 989 | 2 | 6 |
| datasets-in-hr-analytics-applied-ai | nlp | 1000 | 2 | 6 |
| go-to-college-dataset | education | 1000 | 2 | 9 |
| virtual-reality-experiences | english | 1000 | 5 | 5 |
| classification-in-asteroseismology | physics, astronomy | 1001 | 2 | 3 |
| performance-prediction | sports, basketball | 1340 | 2 | 20 |
| minecraft-piracy-dataset | games, law | 1423 | 2 | 2 |
| ibm-attrition-dataset | employment | 1470 | 2 | 12 |
| fake-bills | finance, security | 1500 | 2 | 6 |
| red-wine-dataset | alcohol | 1599 | 6 | 11 |
| pistachio-types-detection | food, nutrition | 1718 | 2 | 15 |
| rms-lusitania-complete-passenger-manifest | military, history | 1961 | 2 | 14 |
| travel-insurance-prediction-data | insurance, travel, business | 1987 | 2 | 8 |
| fetal-health-classification | mortality, health | 2126 | 3 | 20 |
| superstore-marketing-campaign-dataset | marketing, business | 2240 | 2 | 20 |
| room-occupancy | buildings, environment | 2665 | 2 | 4 |
| league-of-legends-stats-s13 | sports | 2683 | 6 | 10 |
| water-potability | environment, energy | 3276 | 2 | 9 |
| nfl-combine-performance-data-2009-2019 | sports | 3477 | 2 | 17 |
| breast-cancer | cancer | 4024 | 2 | 15 |
| tamil-nadu-2021-state-elections | politics, india | 4232 | 2 | 9 |
| patient-treatment-classification | health, business | 4412 | 2 | 10 |
| employee-future-prediction | employment | 4653 | 2 | 8 |
| white-wine-quality | education, alcohol | 4898 | 7 | 11 |
| personal-loan-modeling | finance, banking | 5000 | 2 | 12 |
| sloth-species | animals | 5000 | 3 | 6 |
| gender-classification-dataset | education | 5001 | 2 | 7 |
| stroke-prediction-dataset | health | 5110 | 2 | 9 |
| full-filled-brain-stroke-dataset | health, medicine | 5182 | 2 | 9 |
| wall-following-robot | robotics, health | 5455 | 4 | 24 |
| bundesliga-seasons | football, sports | 5508 | 2 | 16 |
| antarctica-hotpoints-20002021climate-change-nasa | environment, climate | 6127 | 4 | 10 |
| apple-watch-and-fitbit-data | exercise, health | 6264 | 6 | 17 |
| driving-behavior | education, law | 6728 | 3 | 6 |
| spanish-wine-quality-dataset | alcohol, food | 7500 | 8 | 9 |
| water-quality | pollution, energy | 7999 | 2 | 19 |
| ml-marathon-dataset-by-azure-developer-community | finance, software | 8371 | 2 | 16 |

| bank-customer-churn | finance, banking, business | 10000 | 2 | 16 |
|---|---|---|---|---|
| defaulter | finance | 10000 | 2 | 3 |
| paris-housing-classification | housing | 10000 | 2 | 17 |
| predictive-maintenance-dataset-ai4i-2020 | manufacturing | 10000 | 2 | 11 |
| sloan-digital-sky-survey | business, physics, astronomy | 10000 | 3 | 14 |
| oranges-vs-grapefruit | food | 10000 | 2 | 5 |
| predicting-credit-card-customer-attrition-with-m | business, lending | 10127 | 2 | 19 |
| customer-analytics | business | 10999 | 2 | 10 |
| microcalcification-classification | cancer | 11183 | 2 | 6 |
| market-research-survey | marketing | 14898 | 2 | 6 |
| asteroid-impacts | astronomy | 15635 | 2 | 14 |
| world-air-quality-index-by-city-and-coordinates | pollution | 16695 | 6 | 13 |
| 177k-english-song-data-from-20082017 | music | 17734 | 2 | 19 |
| pulsar-classification-for-class-prediction | physics | 17898 | 2 | 8 |
| rice-type-classification | food, business | 18185 | 2 | 10 |
| telescope-spectrum-gamma-or-hadron | astronomy | 19020 | 2 | 10 |
| noaa-atlantic-hurricane-database | environment | 19066 | 9 | 12 |
| hr-analytics-job-change-of-data-scientists | education, business | 19158 | 2 | 12 |
| mental-health-social-media | nlp, health | 20000 | 2 | 7 |
| the-social-dilemma-tweets | internet, business | 20068 | 3 | 11 |
| taxol-drug-resistance-cell-lines-in-breast-cancer | biology, cancer, health | 21312 | 4 | 3 |
| multijet-primary-dataset | physics | 21726 | 6 | 13 |
| fifa-world-cup-2022-tweets | football, nlp | 22524 | 3 | 4 |
| fifa-world-cup-2022 | football, sports | 23921 | 3 | 22 |
| nasa-near-earth-objects-information | astronomy | 24000 | 2 | 11 |
| lumpy-skin-disease-dataset | health | 24803 | 2 | 19 |
| sentiment-and-emotions-of-tweets | nlp | 24970 | 3 | 6 |
| starbucks-locations-worldwide-2021-version | exercise | 28289 | 4 | 13 |
| hackerearth-customer-segmentation-hackathon | business | 31647 | 2 | 16 |
| wine-quality-data-combined | alcohol | 32485 | 7 | 12 |
| api-access-behaviour-anomaly-dataset | websites, programming | 34423 | 4 | 10 |
| hotel-reservations-classification-dataset | travel, hospitality | 36275 | 2 | 17 |
| the-spotify-hit-predictor-dataset | music, internet, business | 41106 | 2 | 18 |
| dataset-aroma-tahu-berfomalin | english, chemistry | 45000 | 2 | 8 |
| hranalysis | education, business, software | 54808 | 2 | 12 |
| smart-grid-stability | energy, infrastructure | 60000 | 2 | 13 |
| smoke-detection-dataset | environment, safety | 62630 | 2 | 12 |
| exploring-risk-factors-for-cardiovascular-diseas | genetics, health | 70000 | 2 | 11 |

## A.3 PROMPT DESIGN

To leverage the language capabilities of large language models and enable them to learn from and make inferences on tabular data, we convert the data into textual form. Our approach employs a unique prompt design, which consists of two main components: feature serialization and task instruction. We use GPT-4 (OpenAI, 2023) to automatically generate meta information for each dataset, combining data analysis with manual corrections. This refined meta information is then used to transform tabular data into textual data. The following sections provide a detailed introduction to the design of feature serialization and task prompts in the meta information 1, with the corresponding prompt example shown in 2.

- **Feature Serialization**: We design two serialization templates catering to distinct feature types: numerical and categorical feature templates. The numerical feature template is composed of a subject, verb, and decimal place, while the categorical feature template integrates a subject, verb, and value dictionary, which associates each category of a feature with its real-world meaning. The primary motivation for this feature template design is to transform each feature, along with its corresponding value, into a fluent sentence. More precisely, the feature prompt adopts the structure "The *subject verb value*".

- **Task Instruction**: We specify the task background, detailing the objective and providing answer choices, which are accompanied by their corresponding label indices as found in the original data. This comprehensive definition ensures a clear understanding of the task and facilitates accurate evaluation of the model's performance.

As a result of this design, we can effortlessly incorporate context samples into the prompt, enabling the model to extract more information from the context for inference. Specifically, we carry out feature serialization on each context sample, concatenate the answer and the feature of the context sample to form a context prompt, and place it before the target prompt that requires prediction. The example prompt with context samples can be found in case study section.

Listing 1: Meta information example

```
1  {
2      "feature_serialization": {
3          "training_hours": {
4              "subject": "number of training hours completed by the
                   ↪ candidate",
5              "verb": "is",
6              "type": "num",
7              "decimal": 1
8          },
9          "company_type": {
10             "subject": "current employer of the candidate",
11             "type": "cat",
12             "verb": "is",
13             "value_dict": {
14                 "Early Stage Startup": "an early stage startup",
15                 "Funded Startup": "a funded startup",
16                 "NGO": "an NGO",
17                 "Public Sector": "a public sector",
18                 "Pvt Ltd": "a private limited company"
19             },
20             "categories": 6
21         }
22     }
23     "task_instruction": {
24         "role_prompt": "You are an expert in HR analytics and data
                ↪  science recruitment.",
25         "task_prompt": "I will provide some features of a
                ↪ candidate, please predict whether the candidate
                ↪ genuinely wants to work for the company after
                ↪ training or is looking for new employment.",
26         "answer_prompt": "Work for the company[1] or look for a
                ↪ new employment[0]?"
27     }
28 }
```

Listing 2: Prompt example

```
You are an expert in HR analytics and data science recruitment.
I will provide some features of a candidate, please predict
    ↪ whether the candidate genuinely wants to work for the
    ↪ company after training or is looking for new employment.
Work for the company[1] or look for a new employment[0]?
Features: The number of training hours completed by the candidate
    ↪ is 6.5.
The current employer of the candidate is a funded startup.
Answer: 1
```

## A.4 CONNECTIONS WITH TABLLM DATASETS

All 115 datasets for GTL have been included in Table 3. We have carefully examined the connections between these 115 datasets and the nine evaluation datasets built by Hegselmann et al. (2023). We find that the general knowledge of two evaluation datasets, Diabetes and Bank, has been covered by some of the datasets for GTL, but there is no data leakage issue. As a result, we can regard the adaptation to these two datasets as a kind of in-domain transferring. While for other evaluation datasets, we do not find explicit connections with the 115 datasets for GTL. Below we include the details on the in-domain transferring datasets to facilitate understanding.

**Diabetes**   Our data collection includes the *early-diabetes-classification* dataset, which is similar to but also have distinct differences with the *diabetes* dataset . A summary of the similarities and differences between these two datasets is as follows:

- **Similarity**: Both datasets address the medical domain, specifically targeting diabetes classification. They share the objective of predicting the presence or absence of diabetes in individuals based on various input features.

- **Differences**:
  - *Target*: The *early-diabetes-classification* dataset emphasizes early diabetes detection, whereas the *diabetes* dataset has a broader scope, targeting diabetes classification without emphasizing early stages.
  - *Features*: The *early-diabetes-classification* dataset comprises 17 features, including age, gender, polyuria, and polydipsia, while the *diabetes* dataset consists of eight features, such as pregnancies, glucose, blood pressure, skin thickness, insulin, BMI, diabetes pedigree function, and age. Apart from the age feature, there is no overlap between the features in these two datasets. In the *diabetes* dataset, all features are numerical, whereas in the *early-diabetes-classification* dataset, the majority of features are categorical, with age being the only exception.
  - *Distribution*: The *early-diabetes-classification* dataset contains 520 records, while the *diabetes* dataset has 768 records. These sample populations may have different demographic characteristics and distributions.

**Bank**   Another group of related datasets consists of the *ml-marathon-dataset-by-azure-developer-community* and *hackerearth-customer-segmentation-hackathon* from our collected datasets, as well as the *bank* dataset from the baseline datasets.

- **Similarities**: All three datasets emphasize customer data and are applicable to a range of marketing and customer-oriented tasks. They encompass demographic and behavioral information about customers, enabling shared domain knowledge that aids in transfer learning.

- **Differences**:
  - *Features*: Utilizing GPT-4 for automatic generation of meta information for each dataset, the feature serialization templates and task instruction prompts exhibit differences across the three datasets. These variations arise from the unique descriptions of the datasets provided on Kaggle.
  - *Data Distribution*: The *ml-marathon-dataset-by-azure-developer-community* dataset comprises 8,371 records, while the *hackerearth-customer-segmentation-hackathon* dataset contains 31,647 records, and the *bank* dataset has 45,211 records. Furthermore, we select a balanced sample of 2,048 shots from each public dataset and ensure no overlap with the *bank* dataset used for evaluation.

## B   IMPLEMENTATION DETAILS

### B.1   BASELINE MODELS

Our study involves two sets of experiments: instruction following and fine-tuning. We utilize different baseline models for each set to comprehensively evaluate the learning methods applied to tabular data.

**Instruction Following.**   We assess the instruction following capabilities of several large language models, dividing them into two categories: black box models and white box models. Black box models include the GPT series, where we can only access the response rather than model parameters and prediction logits. In contrast, white box models, such as T0 and LLaMA, allow us to access the entire model and obtain logits for each class. The descriptions of each baseline model are as follows:

- **GPT-3.5** (Brown et al., 2020): We employ the GPT-3.5-turbo version, which is the most capable GPT-3.5 model optimized for chat and designed for diverse tasks such as natu-

ral language understanding, translation, and summarization. To encourage the model to provide predictions and minimize instances where it refuses to make predictions, we incorporate an additional answer instruction prompt when querying GPT-3.5. This approach allows GPT-3.5 to supply probabilities for predicting specific categories, which can then be used to calculate AUROC. The prompt details can be found in section D.

- **GPT-4** (OpenAI, 2023): A powerful baseline renowned for its top performance in numerous language tasks due to its larger model size and architectural refinements. We also fine-tune the answer instruction prompt for GPT-4. Compared to GPT-3.5, GPT-4 demonstrates superior performance in both zero-shot and in-context learning. Owing to its robust semantic abilities, GPT-4 can respond in a more complex format. We instruct GPT-4 to return predictions that include probabilities for each category simultaneously, with the requirement that the sum of probabilities across all categories equals 1. We give a prompt example in section D.

- **T0 (TabLLM)** (Sanh et al., 2022; Hegselmann et al., 2023): An important baseline that uses a text template provided in the TabLLM repository[5] for the zero-shot setting, generating prompts identical to those in their paper. We reran the zero-shot results using the t-few repository[6]. Since TabLLM does not provide in-context learning prompt templates, we designed our own template for generating prompts with context samples to fairly compare in-context learning capabilities across various LLMs.

- **LLaMA** (Touvron et al., 2023): We employ the LLaMA 2 7B version as our baseline and backbone for generative tabular learning.

**Task-specific finetuning.** For the fine-tuning experiment, we followed the hyperparameter tuning process outlined in TabLLM (Hegselmann et al., 2023) and performed a 4-fold cross-validation on k-shots for both LLaMA and LLaMA-GTL to adjust the hyperparameters. Consequently, our method is comparable not only to the TabLLM method but also to various other deep-learning-based baselines. We have included the results of these additional baseline models in TabLLM:

- **Logistic Regression**: A linear model for estimating binary response probabilities based on predictor variables. Simple, fast, and often used as a classification baseline.

- **XGBoost** (Chen & Guestrin, 2016): An advanced gradient-boosted decision tree algorithm renowned for its efficiency and high performance. Popular for handling diverse datasets and commonly used in machine learning competitions.

- **LightGBM** (Ke et al., 2017): A gradient boosting framework employing tree-based learning algorithms, designed for efficiency and scalability, offering faster training speeds and lower memory usage than other techniques.

- **SAINT** (Somepalli et al., 2021): A deep learning model for tabular data that uses self-attention mechanisms and transformer architecture to capture complex interactions and dependencies.

- **TabNet** (Arik & Pfister, 2021): A neural network architecture for tabular data that employs sequential attention to select relevant features for each decision step, focusing on important features while reducing model complexity.

- **NODE** (Shwartz-Ziv & Armon, 2022a): A deep learning model that utilizes Neural Oblivious Decision Ensembles to learn expressive decision rules, combining decision tree and deep learning strengths for accurate, interpretable predictions.

- **TabPFN** (Hollmann et al., 2023) A deep learning model that combines Positional Feature-wise Networks with transformer-based architectures for tabular data, capturing both local and global feature interactions to enhance performance in various tasks

## B.2 GENERATIVE TABULAR LEARNING (GTL)

We performed specific processing steps in constructing large-scale datasets for generative tabular learning. The following procedures were employed to ensure data quality and diversity:

---

[5] https://github.com/clinicalml/TabLLM
[6] https://github.com/r-three/t-few

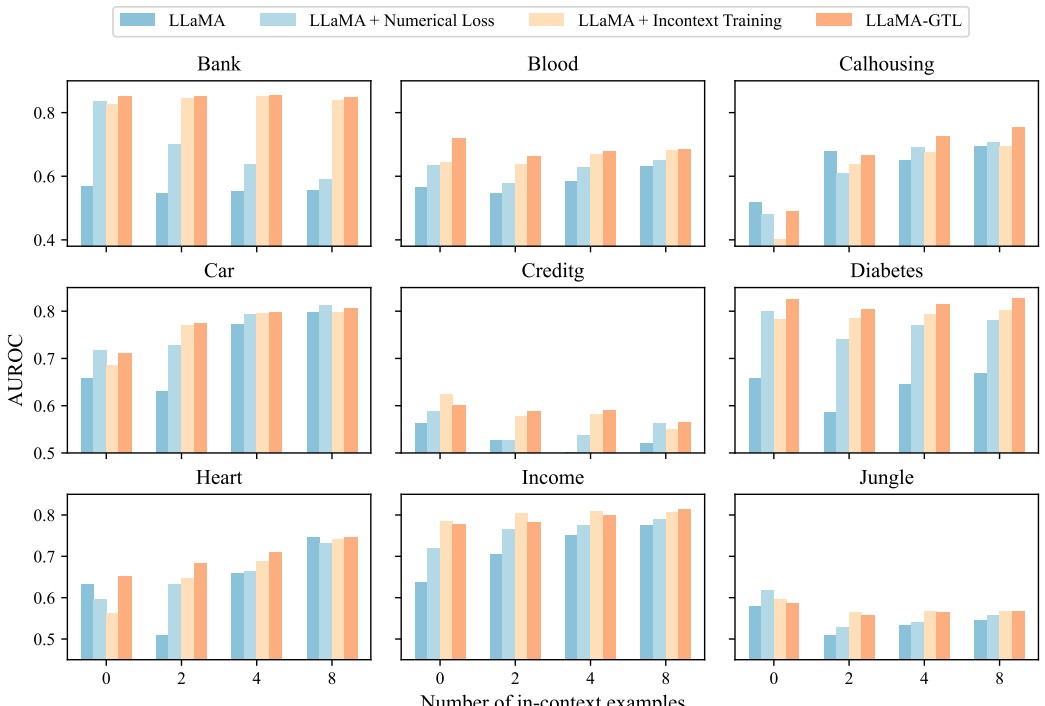

Figure 2: Ablation study of instruction following task. Both the numerical loss and in-context training improves the prediction performance.

- **Feature Filtering**: We filtered out features with no semantic information, such as ID-type features, hash strings, or features with uninterpretable meanings, from each dataset.

- **Balanced Sampling**: To prevent overfitting to a specific domain and ensure data diversity, we limited the training dataset to 2048 samples for each dataset. We attempted to balance the number of samples for each class in classification datasets. However, if a certain class had insufficient samples, we included all available samples from that class without re-sampling.

- **Context Construction**: To enable generative tabular learning to reason and predict based on context, we created samples with contextual information. For each sample, we randomly selected samples from the corresponding dataset as context. We set the number of contextual samples to 2 and 4, ensuring that the target sample's distribution remained unchanged. This approach enhanced in-context learning capabilities without compromising zero-shot performance.

**Hyper-parameters.** We employ the LLaMA 2 7B (Touvron et al., 2023) model as the backbone for our experiments. For generative tabular learning, we utilize a fixed learning rate of 1e-5 and a batch size of 512, without incorporating any scheduler or warmup. The training process includes gradient updating a total of 256 batches with optimizer AdamW (Loshchilov & Hutter, 2018). We set the limitation of maximum token numbers to 4096, which ensures that all samples are within the acceptable range and prevents truncation.

**Experimental Environments.** The TabFM model is implemented using PyTorch version 2.1.0 and executed on CUDA 12.1, running on NVIDIA Tesla A100 GPUs. As for GTL, training is performed on single nodes equipped with 8 A100 GPUs and the micro batch size is 4 for each GPU. With an overall batch size of 512, the gradient accumulation steps amount to 16. In GTL, updating the gradient for 256 batches, which comprises 131,072 samples, takes approximately 2 hours.

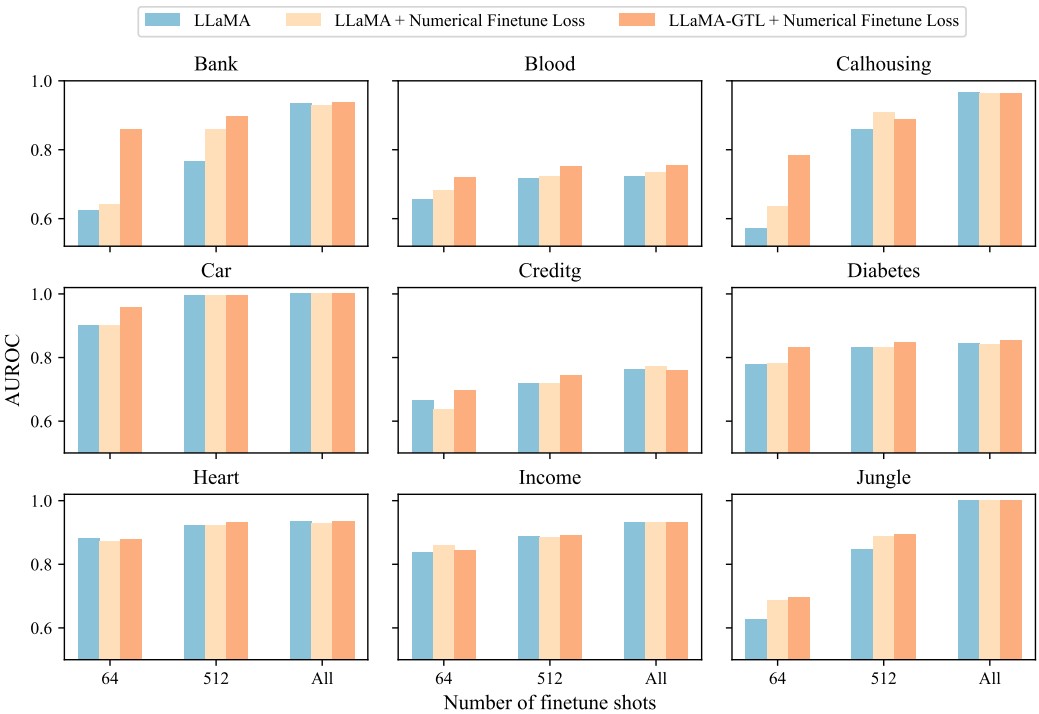

Figure 3: Ablation study of the task-specific finetuning task. The numerical loss also improves the base LLM performance during the finetuning process.

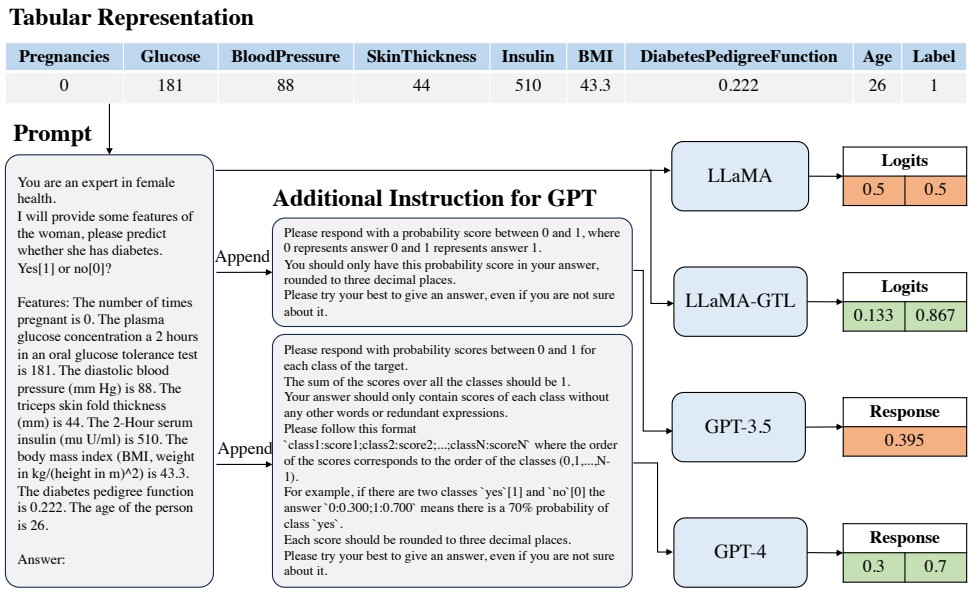

Figure 4: Zero-shot prediction case. LLaMA-GTL significantly improves the tabular knowledge for zero-shot prediction over LLaMA.

**Tabular Representation**

| median_income | housing_median_age | total_rooms | total_bedrooms | population | households | latitude | longtitude | Label |
|---|---|---|---|---|---|---|---|---|
| 3.1875 | 23.0 | 1419 | 261 | 706 | 269 | 38.79 | -121.24 | 0 |
| 6.5891 | 17 | 1625 | 239 | 703 | 224 | 38.29 | -122.15 | 1 |
| 4.9394 | 31 | 2421 | 389 | 1348 | 413 | 33.83 | -117.95 | 1 |
| 1.1607 | 50 | 1050 | 288 | 485 | 260 | 39.51 | -121.55 | 0 |
| 1.2 | 20 | 37 | 11 | 34 | 8 | 35.32 | -119.04 | 0 |

**Zero-shot Prompt**

You are an expert in statistics and housing.
I will provide some features of a block of houses in a California district in 1990. Please predict whether the price of the block of houses is higher than the median of this district or not.
Higher[1] or lower[0]?
Features: The median annual income of households (in tens of thousands dollars) is 1.2. The median age of the houses in years is 20. The total number of rooms is 37. The total number of bedrooms is 11. The total number of people residing within the block is 34. The total number of households within the block is 8. The latitude is 35.32. The longitude is -119.04.
Answer:

| **LLaMA** | |
|---|---|
| 0.392 | 0.608 |

| **LLaMA-GTL** | |
|---|---|
| 0.294 | 0.706 |

**In-context Examples Prompt**

You are an expert in statistics and housing.
I will provide some features of a block of houses in a California district in 1990. Please predict whether the price of the block of houses is higher than the median of this district or not.
Higher[1] or lower[0]?
Features: The median annual income of households (in tens of thousands dollars) is 3.1875. The median age of the houses in years is 23. The total number of rooms is 1419. The total number of bedrooms is 261. The total number of people residing within the block is 706. The total number of households within the block is 269. The latitude is 38.79. The longitude is -121.24. Answer: 0
Features: The median annual income of households (in tens of thousands dollars) is 6.5891. The median age of the houses in years is 17. The total number of rooms is 1625. The total number of bedrooms is 239. The total number of people residing within the block is 703. The total number of households within the block is 224. The latitude is 38.29. The longitude is -122.15. Answer: 1
Features: The median annual income of households (in tens of thousands dollars) is 4.9394. The median age of the houses in years is 31. The total number of rooms is 2421. The total number of bedrooms is 389. The total number of people residing within the block is 1348. The total number of households within the block is 413. The latitude is 33.83. The longitude is -117.95. Answer: 1
Features: The median annual income of households (in tens of thousands dollars) is 1.1607. The median age of the houses in years is 50. The total number of rooms is 1050. The total number of bedrooms is 288. The total number of people residing within the block is 485. The total number of households within the block is 260. The latitude is 39.51. The longitude is -121.55. Answer: 0
Features: The median annual income of households (in tens of thousands dollars) is 1.2. The median age of the houses in years is 20. The total number of rooms is 37. The total number of bedrooms is 11. The total number of households within the block is 34. The total number of households within the block is 8. The latitude is 35.32. The longitude is -119.04.
Answer:

| **LLaMA** | |
|---|---|
| 0.469 | 0.531 |

| **LLaMA-GTL** | |
|---|---|
| 0.867 | 0.133 |

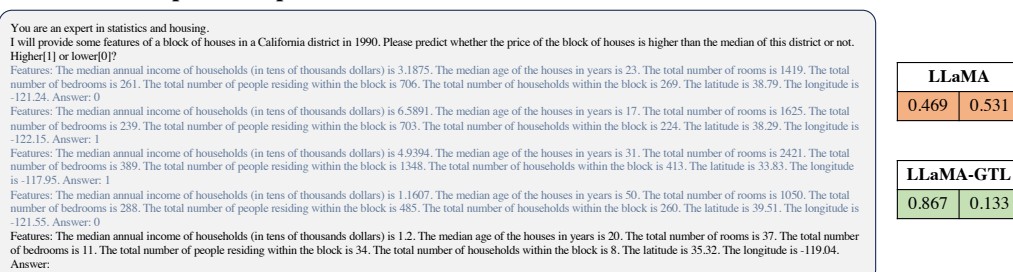

Figure 5: In-context learning case. LLaMA-GTL can effectively learn from the in-context examples.

## C  ADDITIONAL EXPERIMENTS

### C.1  ABLATION OF INSTRUCTION FOLLOWING PREDICTIONS

Here we present the ablation experiments for studying the influence of numerical loss and in-context training in instruction following tasks. We compare the performance of 1) LLaMA-2 7B without pretuning, 2) LLaMA + numerical loss for pretuning, 3) LLaMA + in-context pretuning, and 4) LLaMA-GTL on the nine evaluation datasets, as displayed in Figure 2.

Pretuning with numerical loss improves the zero-shot performance on all datasets except C. Hous. and Heart. LLaMA with in-context training exhibits a similar pattern, underperforms the base LLaMA on C. Hous. and Heart for zero-shot predictions. As the number of in-context shots increases, their performance eventually surpasses LLaMA on C. Hous. and are similar on Heart. The overall superior performance indicates the design of the numerical loss and in-context training is very useful for tabular predictions.

Moreover, LLaMA pretuned with numerical loss but not in-context training shows inconsistent or even decreasing in-context performance during inference. In contrast, LLaMA pretuned with in-context training consistently benefits from more in-context examples and achieves higher AUROC, except on Creditg and Jungle datasets. This indicates that the in-context training is a more important component of LLaMA-GTL.

Finally, LLaMA-GTL that combines the two pretuning components achieves the best overall prediction performance, which confirms the necessity and mutual complementarity of the two designs.

### C.2  ABLATION OF TASK-SPECIFIC FINETUNING PREDICTIONS

Here we present the ablation experiments for studying the influence of employing numerical loss in finetuning process. We compare the finetuning performance of 1) LLaMA-2 7B without pretuning, 2) LLaMA + numerical loss for finetuning, and 3) LLaMA-GTL on the nine evaluation datasets, as shown in Figure 3.

Employing numerical loss in finetuning process lead to superior 64-shot performance on all datasets except small decreases on Creditg and Heart datasets. As more samples are added for training, the numerical loss further boost the prediction performance for 512-shot cases, matching or surpassing the base LLaMA on all datasets. This observation confirms that numerical loss is also useful in the finetuning process.

LLaMA-GTL significantly outperforms the baselines on Bank, Calhousing for 64 shots, indicating that the pretuning stage is essential for few-shot finetuning tasks. When using all instances for training, the models exhibit similar performance, with negligible advantage of LLaMA-GTL on Blood. This indicates that the all the three models reaches the upper bound of LLaMA on these datasets.

## D  CASE STUDIES

Here we provide concrete examples of instruction following task. Figure 4 shows the zero-shot predictions compared with GPT-3.5 and GPT-4. LLaMA and GPT-3.5 make incorrect prediction while GPT-4 and LLaMA-GTL correctly predicts the health condition of the person. Figure 5 displays the in-context learning capability of GTL. Without in-context examples, both LLaMA and LLaMA-GTL fail to evaluate the house value. When provided with in-context samples, LLaMA still struggles while LLaMA-GTL successfully learned to predict the correct label.

