# OpenReview forum: "Towards Foundation Models for Learning on Tabular Data"
_ICLR.cc/2024/Conference — Submitted to ICLR 2024_

### Official Review · Reviewer_2oDN · 2023-10-23

**Soundness:** 2 fair
**Presentation:** 3 good
**Contribution:** 2 fair
**Rating:** 5
**Confidence:** 4

**Summary:**

This paper presents a method to train an LLM-backbone foundation model for tabular data. The premise is that, by including examples from from tabular datasets serialized into text, they can train a model that generalizes well to new tabular datasets in the zero and few-shot setting. They find that their model trained in this way leads to improved generalization on a held out set of tabular tasks compared to tabllm and gpt* models. Additionally, they find their model performs comparably to classic tabular models like LR and xgboost when trained with more data points.

**Strengths:**

- Tabular data is a widely used data format. Improvements in this area are very important and high impact.
- The presentation of the paper is quite clear and easy to follow.
- The pre-training data set seems comprehensive and collects many relevant dataset for tabular pre-training.
- The results in Table 1 are quite promising -- in the few shot setting, the proposed methods either offers clear gains or is comporable to GPT-4 which is exciting

**Weaknesses:**

This paper offers promise in terms of performance gains on several tabular benchmarks but currently leaves quite a few questions unanswered about how LLMs like the trained foundation model might be used in place of current tabular models and what the tradeoffs are in leveraging these techniques.

1. The foundation model is leveraging prior knowledge in the form of feature names + values. What role do these play in generalization to new tasks? If features are anonymous or have values scaled to an unexpected range (common occurances in practice) how much does this hurt generalization?
2. For tabular learning, the presence of (potentially many) noisy / useless features is quite common. Does this hurt generalization of this method, as the model might pay attention to noisy or useless features?
3. For table 2, I'm a bit concerned about the gap between logistic regression and xgboost on these tasks. It seems like these datasets are simple enough that logistic regression matches xgboost performance. In more complicated real world cases, this gap is often quite greater, as the ability of xgboost to fit complex patterns in the data is needed for good performance. So, I'm concerned the evaluation isn't reflective of many real world uses cases. To be fair, in certain domains, such as medical domains, I've seen results demonstrating logistic regression leads to sota performance, but in many other domains this is not the case, so am unsure how much I can expect the utility of this model to generalize.


Other considerations:
- It's a bit strange to fit the baselines (LR, lightgbm, and xgb) in table 2 with no (?) hyperparameter tuning -- I didn't see a description of this process here. E.g., even with only 64 data points, its often possible to get considerably better results by tuning on LOO or k-fold CV on the training set for instance and considering a couple different hyperparameter combinations. Without this, I'm a somewhat suspicious the true performance of these baslines is a point or two higher than currently presented.

There's also an additional related work here which could be a useful benchmark: https://arxiv.org/abs/2304.13188

**Questions:**

See weaknesses

---

> ### Author Response · Authors · 2023-11-23
> **Response to Reviewer 2oDN**
>
> We greatly appreciate your recognition of the promising direction and results of our work. Your insightful comments indeed highlight several situations that could influence the generalization of our methods. These are issues we have strived to address in our practical implementation, but perhaps we did not elaborate sufficiently in the paper. We will now address each of your concerns in detail and welcome further suggestions and discussion.
>
> **Response to Weakness 1**
>
> Thank you for raising this insightful issue.
> 1. Feature names and values indeed serve as bridges between tabular data and language. Without the semantic information provided by feature names and values, we cannot effectively convey the necessary information to LLMs, thus impeding their ability to leverage prior knowledge when transferring to new tasks. However, even in the absence of meaningful feature names, the statistical learning ability of LLMs enhanced by GTL can still be utilized in few-shot instruction following and fine-tuning scenarios.
> 2. In practice, we can adopt several methods to express values that fall into unexpected ranges, thus partially avoiding such occurrences. For instance, we can **add units to scale the numerical values to a normal range, or use scientific notation to express extremely large or small values**.
>
> Notably, we found that specifying the correct feature unit is important.
> For example, in public datasets collected from Kaggle, height and weight are often essential features for predicting targets such as BMI or certain diseases. However, the unit for those features can differ across datasets. In bmidataset [1], the unit for height is cm and for weight is kg. But in predict-diabetes-based-on-diagnostic-measures dataset [2], the unit for height is inches and for weight is pounds.
> Specifying the correct unit for each dataset helps to avoid distribution mismatches and misunderstandings for LLMs. This principle applies to other features like currency units (from dollar to EUR) and time units (from seconds to years).
>
> [1] https://www.kaggle.com/datasets/yasserh/bmidataset
>
> [2] https://www.kaggle.com/datasets/houcembenmansour/predict-diabetes-based-on-diagnostic-measures
>
> **Response to Weakness 2**
>
> We appreciate your thoughtful emphasis on the potential impact of noisy and irrelevant features. While these methods intuitively improve generalization, due to limited resources, we have not yet conducted quantitative experiments to measure the impact of these factors specifically. But we would like to share how we try to address this issue:
>
> 1. When constructing high-quality training data, we remove certain useless features such as ID numbers or hash values. This helps to reduce the amount of noise in the data.
> 2. During the training process, we randomly drop a small percentage of the features. This strategy prevents the model from focusing excessively on one or a few features, thus promoting more balanced attention distribution across all relevant features.
>
> **Response to Weakness 3**
>
> Thank you for raising your concern about the evaluation datasets. Our current evaluation followed by TabLLM may indeed be relatively simple, contributing to the observed performance similarities. We would like to expand our evaluation to include a wider variety of heldout datasets and tasks from collected tabular data corpus. This will allow us to provide a more comprehensive view of the performance characteristics and generalizability of our model.
>
> **Response to your considerations**
>
> Thank you for your question regarding the tuning process for the baselines. In fact, we did conduct hyperparameter tuning for all baselines. As detailed in the "Task-specific finetuning" section in Appendix B.1, we followed the hyperparameter tuning process outlined in TabLLM. This process involves **performing hyperparameter tuning based on 4-fold cross-validation on k-shots**. We apologize for any confusion caused by the lack of clarity in our paper and appreciate your diligence in pointing out this issue.

---

### Official Review · Reviewer_zaKi · 2023-10-31

**Soundness:** 3 good
**Presentation:** 3 good
**Contribution:** 2 fair
**Rating:** 3
**Confidence:** 4

**Summary:**

This paper tackles the problem of building tabular foundation models, or neural networks that can operate with high performance across several downstream tabular tasks. The particular approach is to fine tune pretrained LLMs to better handle tabular tasks. In a range of experiments the tactic is examined in the zero shot, in-context learning, and fine-tuning settings.

**Strengths:**

1. To my knowledge, this paper is original -- the idea of fine tuning LLMs like Llama for general purpose tabular tasks is novel.
2. The quality and clarity of the writing are good.

**Weaknesses:**

1. The central results in Table 1 are unclear. I think this is in part due to the massive multi-row table and in part due to the hazy conclusions.
    1. For example, on the Bank data, the only method that ever achieves AUC of 85 is LLaMA-GTL but it does this with no examples (#contexts = 0). Suggesting that LLaMA-GTL is great for this task, but in-context learning doesn't help it.
    2. On C. Hous. and Car datsets, LLaMA-GTL strictly underperforms baselines with 0 contexts.
    3. On Diabetes, Heart, Income, and Jungle, LLaMA-GTL again shows no better performance than 0-context baselines

--> This tells a less-than compelling story about when LLaMA-GTL is the right tool. Seemingly for free (no context, no training) existing methods are more general. I do see that the GTL component provides LLaMA with a significant boost in almost all cases.

2. The  results presented in Table 1 are a weakness of the paper also because the claims made following Table 1 include (i) in-context learning doesn't help; (ii) GTL helps in most zero-shot settings; (iii) GTL helps in few-shot in-context settings; Point (i) is made clearly by the table, but strikes me as a limitation -- foundation models tend to do better with in context examples, if TabFMs don't this needs to be investigated further. Points (ii) and (iii)  are funny comparisons since the number of in-context examples is presented as a control variable, but GTL models have a bunch more training. This may be a strength of GTL models over base models, but this is simply an example of where fine tuning base models makes them better at specific tasks and therefore feels like a less than novel finding.

3. The results in Table 2 are similarly hard to interpret. See the questions below.

Minor issues not affecting my score:
1. The comparison of the proposed method to the baselines and in various settings are presented in massive and complicated tables. It took me a very long time and repeated readings to follow the rows and columns and draw conclusions. This is a relatively minor point, as I acknowledge that the data itself is there, but I suggest smaller tables or plots/figures for some of the experiments to help the reader draw conclusions and interpret the results faster.
2. the indices in Section 3 are quite confusing and difficult to follow. I'm fairly certain I understand what goes into these models but these equations in their current form confused rather than clarified the details for me.
3. `We employ LLaMA-2-7B (Touvron et al., 2023) as our base LLM, which will be simply referred to as ’LLaMA’ in the following sections for brevity. Additionally, we denote the obtained TabFM after the GTL stage as LLaMA-GTL` looks like it's repeated on Page 6.

**Questions:**

1. How is AUC computed with language models? I see in Appendix Figure 4 that there are logits, but I don't follow exactly how they are computed. Can the authors elaborate?
2. How does GTL work when paired with LLMs other than LLaMA? A little more breadth here could make the story more compelling. I'm sensitive to limited compute resources, but I also think that more than one model would help illuminate when/where GTL can be effective.
3. Can the authors add any intuition or explanation about when TabFM's might be succeeding/failing? For example, do the datasets on which it works well have anything in common (number of features, categorical/numerical features, number of classes, similarity to datasets in the GTL training sets, etc)? Without any discussion or hypotheses around this point, I think the results in this paper seem fragmented and difficult to parse.

I look forward to discussing these questions with the authors and the other reviewers.

---

> ### Author Response · Authors · 2023-11-23
> **Response to Reviewer zaKi (Part 1/2)**
>
> We sincerely appreciate your recognition of the novelty and contribution of our work. We agree that while there is still a journey ahead for us to fully realize a tabular foundation model, the direction is indeed promising. We apologize if certain aspects of our work were not clearly articulated. We will address each of your concerns and engage in a detailed discussion to provide further clarification.
>
> **Response to Weakness 1**
>
> Thank you for highlighting this observation.
> 1. We pointed out in our paper that **"Introducing in-context examples does not necessarily lead to performance improvements over zero-shot inference in new tabular tasks"** (In section 4.1 Zero-shot vs In-context), which is not ambiguous. We believe this conclusion provides valuable insights about the existing conflict between general knowledge and statistical knowledge derived from a few in-context examples. Finding a trade-off between these two types of knowledge in current LLMs is indeed challenging.
>
> 2. When utilizing LLaMA-GTL as a tool, we recommend conducting zero-shot prediction when the task only involves common knowledge. Conversely, for domain-specific tasks or those requiring statistical learning, providing some examples for in-context learning would be more appropriate. Furthermore, in practice, an ensemble approach could be adopted to combine zero-shot and in-context learning for the final prediction. We are open to further discussion and welcome any suggestions you may have.
>
> **Response to Weakness 2**
>
> 1. We would like to reiterate our first conclusion, which is that the effectiveness of in-context learning is largely dependent on the specific data and task at hand. Although the effectiveness of zero-shot and in-context learning varies with different datasets, it is clear that both capabilities are indispensable.
>
> 2. With GTL, the zero-shot and in-context learning abilities of LLM are both enhanced. Specifically, pretraining on large-scale diverse data enhances LLM's common knowledge on tabular tasks, improving its transferability to unseen datasets. Learning from data with context samples enhances LLM's ability to gain statistical knowledge from a few examples. For instance, on the C.Hous dataset, there is no performance gain for GTL in zero-shot settings compared to LLaMA, but with contexts=4 and 8, there is a notable improvement.
>
> 3. It is possible that in-context learning could have more stable effects through the use of more powerful models or more sophisticated prompt designs. We are working towards this goal and welcome further discussion and exchange of ideas.

---

> ### Author Response · Authors · 2023-11-23
> **Response to Reviewer zaKi (Part 2/2)**
>
> **Response to Question 1**
>
> 1. For black box models, such as GPT3.5 and GPT4, we add answer instructions to the prompt, which requests models to respond in a fixed answer format (The additional instructions can be found in Figure 4 in Appendix).
>
> 2. For white box models like LLaMA, we obtain logits for each class by accessing the model outputs after the [answer begin token] and applying a softmax function over all categories, which is then used to calculate the AUC.
>
> **Response to Question 2**
>
> Thank you for your suggestion. Due to limited computational resources, we have only experimented with LLaMA-7B so far. However, we are actively exploring the extension of GTL to larger and more powerful LLMs.
>
> **Response to Question 3**
>
> We believe that the success or failure of TabFM's is related to various factors. Here are a few aspects:
>
> 1. **The similarity to GTL training sets** can lead to success or failure. The similarity lies in two aspects, including domain knowledge and task knowledge. A successful case includes the Diabetes and Bank datasets, which have similar domain knowledge to our training sets (details are clarified in Appendix A.4). A failure case is observed on the Jungle datasets, which require task-specific knowledge. All LLMs failed in both zero-shot and few-shot context scenarios. However, after finetuning, several traditional models and LLMs could achieve an AUC close to or equal to 1. This guides us to construct richer and more diverse training data.
>
> 2. **Data distribution** may also have an impact. For example, the C.Hous dataset includes features such as longitude and latitude, which are strongly correlated with the target house value. For instance, nearly all sample values for latitude lie between 37.5 and 37.9. As the latitude distribution in other datasets in the training data is more sparse and not concentrated in this range, LLMs are not sensitive enough to these values, resulting in poor zero-shot performance. However, once context examples are provided, the model can quickly find the association between the feature and the target based on its statistical learning ability. This is particularly true for LLaMA-GTL, which performs well in in-context learning scenarios.

---

### Official Review · Reviewer_RmY6 · 2023-10-31

**Soundness:** 2 fair
**Presentation:** 2 fair
**Contribution:** 3 good
**Rating:** 3
**Confidence:** 5

**Summary:**

The paper proposes a method called Generative Tabular Learning (GTL) for transforming pretrained large language models (LLM) into foundational models for tabular data problems (TabFM). In other words: `TabFM = PretrainedLLM + GTL`. A given LLM trained with GTL can be applied to new unseen tabular tasks in two ways:
- immediately following instructions in zero-shot and in-context regimes
- or after additional finetuning on the task data

The main components of GTL:
- data: a collected set of 115 public datasets with (available or generated by GPT-4) task and column descriptions.
- the loss is based on:
    - traditional language modeling where tabular objects are represented as token sequences
    - feature reconstruction

The main claims:
- *"This approach endows TabFMs with a profound understanding and universal capabilities essential for learning on tabular data"*
- *"excel in instruction-following tasks like zero-shot and in-context inference"*
- *"achieves remarkable efficiency and maintains competitive performance with abundant training data"*

**Strengths:**

- The collected set of 115 datasets is a valuable contribution.
- Modulo the data collection, the method itself is simple (in a good way).
- (Table 1) Promising results in the zero-shot and few-shot regimes, where the proposed LLaMA-GTL is clearly better than the vanilla LLaMA.
- I greatly appreciate the appendix in general, and section A.4 in particular. This transparent analysis is a big plus to me.
- The story is mostly easy to follow.

**Weaknesses:**

*I am ready to review the changes and raise my score even if the changes will be significant (though here I can speak only for myself). In a nutshell, my main recommendations are to narrow the scope and allocate significantly more space for detailed communication of limitations and missing analysis. Regarding the priorities, to me, the points 1, 2, 4, 5 (no experiments required) are more important than 3 (requires experiments). That said, in my opinion, the paper can significantly win from addressing 3.*

**(1. How to apply the proposed model to new tasks?)** *TL;DR: in my opinion, this is an important question that deserves more discussion (and, perhaps, it should be positioned as a limitation).*

From Section A.3 of the appendix, my impression is that if I, as a researcher or a practitioner, want to use the proposed method as a baseline, I need:
- *"use GPT-4"*
- do *"manual corrections"*
- *"specify the task background"*

The above list looks relatively demanding (compared to traditional models) and giving a lot of room for making performance significantly worse/better without any intentions of doing so. In that regard, TabLLM looks easier to use. I think this aspect deserves more discussion.

**(2. Positioning, story, claims)** *TL;DR: In my opinion, the proposed method and reported results are significantly more limited that can be expected from the title/abstract/claims/story. I suggest making the communication of the scope more precise starting from the title and abstract.*

My first impression was that the paper proposed something very general and powerful (*"Towards **Foundation** Models"*, *"we propose Tabular **Foundation** Models"*, *"TabFMs with a **profound understanding and universal capabilities**"*, *"our model achieves **remarkable efficiency** and maintains **competitive performance with abundant training data**"*, etc.). In particular, I expected that the proposed method was tested against things like gradient-boosted decision trees and modern tabular DL architectures without any significant limitations and according to the standards and datasets established in the corresponding field (e.g. see `[1]` for one of the latest examples where such models are compared against each other). However, in the "All" section of Table 2:
- The number of tasks: 9
- The number of regression tasks: 0
- The maximum dataset size: ~50K objects
- The number of trivial tasks (solvable with 100% accuracy): 2
- Among the remaining 7 tasks, the number of simple tasks (Logistic regression performs well): 3
- The proposed LLaMA-GTL vs. the vanilla LLaMA: 2-wins/6-ties/1-loss
- The number of tasks where the proposed method is the only best solution: 2
- (Less of an issue in the light of the above, but important for making strong claims about the non-few-shot regime) I recommend using more powerful DL baselines, e.g. see `[1]`.

Additionally, the proposed method implies running a large 7B model, annotating tasks and columns, and limiting the number of features to fit in the context size. Based on the above, in my opinion, it is too early to promote the new method in the non-small-data scope.

Personally, I would focus on the positive things: the promising results from Table 1 and the collected dataset. And, based on the reported results, I recommend changing the title, abstract and claims accordingly, so that the terms "few-shot" and "classification" are explicitly communicated starting from the title. To me, a great example of a transparent communication is the paper "TabPFN: A Transformer That Solves Small Tabular Classification Problems in a Second" by Hollmann et al., where all important things are described in the title and the abstract.

**(3. Methodology)** *TL;DR: the paper introduces many significant changes compared to prior work (e.g. TabLLM): a new backbone, a non-trivial prompt template, language modeling loss, numerical loss. It may be hard to understand what are the important elements of the scheme.*

- Is it necessary for the proposed method to applied to a relatively powerful 7B LLM? Was the T0 model used by TabLLM a bottleneck?
- Prompts are known to have significant effect on the performance of language models `[2]`. In `[3]`, it was shown that the trivial "List" format is already good enough (perhaps, except for the zero-shot regime). Again, this work diverges from the prior work and uses rich prompts insteads. It is unclear whether this is important.
- Additionally, the benchmark from `[3]` may be enough to illustrate the idea of applying LLMs to tabular data for the first time, but overall, to me, it seems to be limited as I explained above. I believe that, in future, the benchmark should be extended. Works like `[1]` and `[4]` may be a source of additional datasets.

**(4. Conceptually important claims)**. In my opinion, the following claims may worth revisiting or clarifying.

*> "Besides, using a text representation enables the easy integration of **crucial** meta information that can hardly be utilized by traditional studies for learning on tabular data, such as feature meanings and background knowledge."*

The quote (implicitly) suggests that the inability of "traditional" models to process textual information is worth addressing. However:
- If a model requires task and feature descriptions to be presented, **it is a limitation** preventing from applying the model to non-annotated tasks. Traditional models are free from this limitation.
- This can be a problem on my side, but I am not aware of studies showing that an average tabular problem contains non-trivial amount of helpful (in terms of task metrics) information in its description. My guess is that it may be true for tiny tasks, but it becomes increasingly non-obvious for larger tasks.

*> "it also showcases performance that approaches, and in instances, even transcends, the renowned yet mysterious closed- source LLMs like GPT-4"*

I recommend providing a relevant citation to support positioning GPT-4 as a strong baseline for the considered scope. Another option is to remove this additional accent on GPT-4 (using it as a baseline is totally fine).

**(5. The "Limitations" paragraph)**. Currently, this paragraph rather discusses future work than limitations. I recommend adding limitations of the method and of the conducted analysis in this paragraph, and move the ideas for future work into a new "future work" paragraph.

**References**

- `[1]` "TabR: Tabular Deep Learning Meets Nearest Neighbors in 2023", Gorishniy et al.
- `[2]` "Calibrate Before Use: Improving Few-Shot Performance of Language Models", Zhao et al.
- `[3]` "TabLLM: Few-shot Classification of Tabular Data with Large Language Models", Hegselmann et al.
- `[4]` "Why do tree-based models still outperform deep learning on tabular data?", Grinsztajn et al.

**Questions:**

Please, see the weaknesses.

---

> ### Author Response · Authors · 2023-11-23
> **Response to Reviewer RmY6**
>
> We are deeply grateful for your thorough review and constructive feedback on our work. We appreciate your expert insights in the domain of tabular learning and find your suggestions regarding our writing to be invaluable. We will certainly strive to improve and refine our work based on your recommendations. We would now like to engage in a discussion on several points you've raised.
>
> **Response to Weakness 1**
>
> 1. In actual applications, **our model can accept free-form input**. The purpose of constructing this data is to enable the model to use high-quality data during training, bringing it as close as possible to natural language processing.
>
> 2. In fact, **TabLLM also requires a significant amount of manual correction during the serialization process**. For instance, each column name and feature value are manually mapped to values that are more similar to natural language. We leveraged GPT-4 to assist us in generating this information due to the richness and volume of our dataset.
> We acknowledge that implementing our method may be more demanding compared to traditional models, but we believe that the additional effort required are necessary towards tabular foundation models.
>
> **Response to Weakness 2**
>
> We appreciate your suggestions regarding the title and claim. However, it's crucial to clarify that our goal is indeed oriented **towards** foundation models, and our preliminary results have been promising. We hope our findings can bring some inspiration to this research field.
>
> **Response to Weakness 3**
> 1. We chose to use the LLaMA 7B for the following reason. Compared to the T0 tokenizer used by TabLLM, LLaMA tokenizer **separates each digit of the numerical value as a token**, offering better generalization. This provides a more unified and flexible approach to numerical encoding and decoding.
>
> 2. We agree with your observation regarding the significance of prompts. We discovered that prompt formatting significantly influence the performance of LLaMA, T0, and GPT. The impact of prompt formatting is evident in both the serialization of a single sample and the additional instruction for context samples. That is why we opted for a prompt format closer to natural language to enhance generalization and transfer capabilities.
>
> 3. We agree with you that the limitations of the data in TabLLM. We are eager to include more additional datasets for a comprehensive evaluation. Your insights on extending the benchmark and the suggestion of potential data sources are greatly appreciated. We will consider to incorporate these valuable data into our future work.
>
> **Response to Weakness 4**
>
> 1. Indeed, for tasks lacking semantic information, the general prior knowledge of LLMs may not be helpful. This can be viewed as a challenge rather than a limitation. Therefore, we need LLMs to have in-context learning and fine-tuning capabilities. These abilities require data knowledge and statistical learning, areas where current LLMs fall short. The comparison between LLaMA-GTL and LlaMA shows promising improvements brought by our method, and we hope this can inspire future research in this field.
>
> 2. Thank you for your suggestion. We recognize GPT-4 as a general and widely-used model that serves as a baseline in many domains. However, as GPT-4 is a closed-source, black-box model with undisclosed training data and technical details, it has an element of mystery to it. Through our work, we aim to improve upon existing open-source efforts, particularly in the field of tabular learning, to approach or even surpass the performance of such closed-source general models. Our intent is to help progress the research in this domain.

---

### Official Review · Reviewer_rfx2 · 2023-11-01

**Soundness:** 1 poor
**Presentation:** 3 good
**Contribution:** 2 fair
**Rating:** 3
**Confidence:** 4

**Summary:**

The paper proposes a method to adapt pretrained large language models (LLMs) to tabular data problems. The proposed adaptation method uses generative modeling of table rows encoded as text with column and task descriptions, and additional loss for reconstructing continuous features. The proposed method applied to the LLaMA 2 7B is tested on the benchmark from prior work on LLMs for tabular data in zero-shot, few-shot and fine-tuning settings.

**Strengths:**

- The goal of creating tabular foundational models is noteworthy and interesting.
- Numerical loss during finetuning seems to help in adapting language models to numerical data.
- The idea of using next token prediction / generative modeling for adapting text models to tabular data is clear and interesting

**Weaknesses:**

- There is a possibility that modern well-trained LLMs like LLaMA memorized the datasets being used as the benchmark in this paper. Almost all datasets were present on the internet (some plentifully) before the knowledge cutoff for those models. Additional experiments on newer datasets or explicit discussion and testing for memorization are necessary to claim remarkable zero-shot performance on tabular tasks.
- Looking at Table 1, the results vary significantly depending on the number of shots (for the same model: see GPT-4 on Diabetes for example), knowing that LLMs are very sensitive to prompting (even prompt formatting `[1]`). Some standard deviation over shot-selection or prompt formats is needed in this table to make conclusions.
- The baselines that propose foundation models for tabular data without LLMs are discussed in related work, but not compared against in the few-shot experiments, I believe this is an important comparison for a paper to claim a "comprehensive comparison".


**References**
- `[1]` Zhao, Zihao, et al. "Calibrate before use: Improving few-shot performance of language models." International Conference on Machine Learning. PMLR, 2021.

**Questions:**

- Could you provide evidence for or against a hypothesis that LLMs (in this case) LLaMA 2 7B memorized popular datasets from the benchmark and the GTL procedure helps with extracting those memorized samples and not using "general knowledge"?
- How does prompt formatting influence zero/few-shot performance?
- How does GTL on top of LLaMA compare to baselines that do not use LLMs, and just use pretraining on multiple tables instead like `[1]` or `[2]`?

Other remarks:
- NODE was introduced in another paper `[3]`

**References**
- `[1]` Yang, Yazheng, et al. "UniTabE: Pretraining a Unified Tabular Encoder for Heterogeneous Tabular Data." arXiv preprint arXiv:2307.09249 (2023).
- `[2]` Zhu, Bingzhao, et al. "XTab: Cross-table Pretraining for Tabular Transformers." arXiv preprint arXiv:2305.06090 (2023).
- `[3]` Popov, Sergei, Stanislav Morozov, and Artem Babenko. "Neural oblivious decision ensembles for deep learning on tabular data." arXiv preprint arXiv:1909.06312 (2019).

---

> ### Author Response · Authors · 2023-11-23
> **Response to Reviewer rfx2 (Part 1/3)**
>
> We are deeply grateful for your acknowledgment of the unique contributions made by this work. Your recognition of our work as both noteworthy and interesting is highly appreciated. In what follows, we would like to have a discussion on your remaining concerns and questions.
>
> **Response to Weakness 1 and Question 1**
>
> Thank you for your insightful question. We agree with your assertion that there is a potential for LLMs to memorize public datasets. This hypothesis is indeed plausible, as corroborated by our experiments, as detailed in Table 1. In zero-shot experiments, we observed notably high AUROC values for various models across different datasets. For instance, GPT4 exhibited a strong performance on the Diabetes and Heart datasets, while T0 performed exceptionally well on the Car and Income datasets.
>
> However, in the case of LLaMA, the performance across all nine evaluation datasets was below 0.7. More specifically, the scores for the Bank and Diabetes datasets were even lower, at 0.57 and 0.66 respectively. These results suggest that LLaMA has neither learned the distribution of these datasets nor memorized them.
>
> Substantially, one of our major findings is that "LLaMA-GTL significantly improves LLaMA in most zero-shot scenarios," particularly on the aforementioned two datasets. This improvement underscores the knowledge transfer capability of generative learning, which allows LLMs to enhance their performance without data memorization.

---

> ### Author Response · Authors · 2023-11-23
> **Response to Reviewer rfx2 (Part 2/3)**
>
> **Response to Weakness 2 and Question 2**
>
> We appreciate your keen observation on the influence of **shot selection and prompt formatting** on model performance. We concur with your insight and would like to delve deeper into our findings in these areas.
>
> On one hand, as highlighted in Section 4.1 of our paper, we observed that **"few in-context examples do not provide robust statistical knowledge",** which underscores the importance of shot selection. We have investigated several shot-selection strategies and noted that LLMs' predictions for a specific sample are sensitive to the chosen shots. To better align with real-world scenarios and enhance the stability of our experimental results, we have adopted a strategy of randomly selecting context samples for each test sample.
>
> On the other hand, our findings indicate that prompt formatting can also significantly influence the performance of LLaMA, T0, and GPT. For instance, in zero-shot setting, through very few feature prompts modifications, performance of TabLLM will decrease a lot. **All of those LLMs without GTL demonstrated high sensitivity to prompt formatting in our evaluations**. The impact of prompt formatting is evident in both the serialization of a single sample and the additional instruction for context samples.  We aimed to explore the potential of each LLM by employing diverse versions of prompt formats, as depicted in Figure 4.

---

> ### Author Response · Authors · 2023-11-23
> **Response to Reviewer rfx2 (Part 3/3)**
>
> **Response to Weakness 3 and Question 3**
>
> Our approach aims to deliver not only task-specific finetuning but also two additional capabilities that are equally crucial: transferring knowledge to new tasks for zero-shot predictions, and quickly adapting to data-specific knowledge with few-shot examples via in-context learning. While the mentioned baselines, XTab and UniTabE, also conducting pretraining on large-scale data, they are more focus on fine-tuning. Our method can simultaneously demonstrate all three capabilities.
>
> More specifically:
>
> We appreciate UniTabE [1] for their effort in collecting a 13B dataset and their unique architecture design, which strengthens the association between column names and values. In the finetuning scenario, it shows promising results over many baselines across seven public datasets. However, there are a few concerns:
> 1. **UniTabE did not make comparisons with other LLMs in the zero-shot scenario**, only comparing their method with random initialization and their own finetuned model, which does not provide meaningful insights.
> 2. In Table 4 of their paper, **UniTabE only displayed the accuracy metric in the zero-shot scenario, which differs from the AUC metric** used in Table 3. This might not be an effective way to evaluate model performance, especially for imbalanced datasets.
> 3. While UniTabE claims to utilize free-form prompts for tuning pretraining and finetuning, their **1-layer LSTM decoder lacks the language ability and prior knowledge compared to LLMs**. This might limit their ability to support free-form questions in downstream tasks without finetuning on the exact same task.
>
> As for XTab [2], they utilize carefully designed featurizers and training objectives for cross-table pretraining, enhancing learning speed and performance. However:
> 1. Their data-specific design might not be applicable to zero-shot and in-context learning scenarios.
> 2. The improvement their method offers over the FT-Transformer baseline is not particularly significant **in heavy finetuning setting** with a win rate of less than 60% over all tasks.
>
> In light of these points, we believe that our method offers distinct advantages. We appreciate your insightful question and will consider including such comparisons in our future work to provide a more comprehensive evaluation.
>
> [1] Yang, Yazheng, et al. "UniTabE: Pretraining a Unified Tabular Encoder for Heterogeneous Tabular Data." arXiv preprint arXiv:2307.09249 (2023).
>
> [2] Zhu, Bingzhao, et al. "XTab: Cross-table Pretraining for Tabular Transformers." arXiv preprint arXiv:2305.06090 (2023).

---

### Meta-Review · Area_Chair_1c2f · 2023-12-06

**Metareview:**

A novel method for training LLM-based foundation models for tabular data is proposed, showing promising initial results in the few-shot setting. However, limitations exist regarding interpretation, generalizability to unseen data with diverse features and noisy data, and evaluation. Further research is needed to address concerns about interpretability, limitations, generalizability, and evaluation before it can be considered impactful. The paper is not suggested to be accepted by reviewers.

**Justification For Why Not Higher Score:**

- Experimental validation overall is seen as insufficient. The proposed claims should be demonstrated in more convincing ways on real-world data, with apples-to-apples comparisons to compelling alternatives. There are concerns about the existing experimental setup as well.
- Paper writing is not strong in conveying the key contributions.
- Generalizability of the method as a foundation model is not well justified.

**Justification For Why Not Lower Score:**

- Fine-tuning large language models (LLMs) for tabular tasks is a new approach.
- Initial results show promise in the few-shot setting, matching or exceeding GPT-4 in some cases.
- The pre-training dataset is comprehensive and relevant for tabular tasks.

---

### Decision · Program_Chairs · 2024-01-16

Reject